# MorphoGen: Evolving Robot Morphologies with Large Language Models

## Abstract

Designing high-performing robot morphologies is a grand challenge for developing specialized autonomous agents. However, the vast, combinatorial, and non-differentiable nature of the morphological design space has been a primary obstacle. Existing methods tackle this problem *indirectly*, relying on either semantically-blind genetic operators or reinforcement learning with predefined modification actions, both of which constrain exploration. In this work, we introduce **MorphoGen**, a novel framework that reframes morphological design as a code generation problem. MorphoGen leverages large language models (LLMs) to *directly* iterate the XML files as codes that define an agent's morphology, solving the original open problem without being limited by any prior constraints or fixed action spaces. Gradient-like textual guidance is provided to steer the evolution of robot morphologies through prompted mutations and crossovers. Our approach allows the LLMs to apply its understanding of structure and syntax to generate complex and semantically coherent design variations, enabling an unconstrained and efficient exploration of the design space. On a suite of challenging locomotion benchmarks, MorphoGen discovers novel and high-performing morphologies, significantly outperforming strong baselines by over 52.9% in downstream motoring evaluation. Our work unlocks a new paradigm for automated robotic design, demonstrating the effectiveness of LLMs in navigating complex, structured engineering search spaces. Codes for our work are released anonymously at https://anonymous.4open.science/r/MorphoGen-ACC/.

## 1 Introduction

The functionality of a robotic agent is fundamentally determined by its physical form, or morphology (Yuan et al., 2022; Matthews et al., 2023). This morphology is often defined by a structured file, such as the XML format used by MuJoCo (Todorov et al., 2012), which serves as the robot's genotype specifying the precise configuration of skeletons, joints, and their corresponding attributes. Automating the search for an optimal genotype is a highly valuable endeavor, as manually designing and adapting robot morphologies for different tasks is a laborious, intuition-driven process. However, this automation is a non-trivial challenge. The primary difficulty lies in the immense, combinatorial design space; with a nearly infinite number of possible XML definitions, it is im-

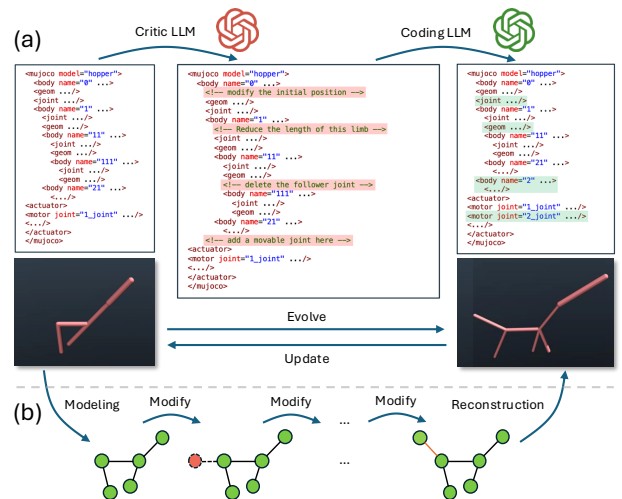

Figure 1: Comparison between paradigms of robot morphology design. (a) The proposed MorphoGen framework, in which LLMs directly operate on the raw XML representation to generate complete robot morphologies in a single step. (b) The conventional approach, which relies on simplified intermediate models and requires multiple handcrafted modification steps.

possible to exhaustively iterate through all potential skeletal structures and attributes. This challenge is compounded by the high computational cost of evaluation, as each candidate morphology necessitates a costly "inner loop" of control policy optimization to assess its fitness.

To manage this complexity, existing methods avoid operating directly on raw XML definitions, instead introducing an intermediate graph-based abstraction as a mediator. In this common paradigm, a robot's morphology is represented as a directed graph where topological elements (nodes and edges) denote body parts, joints, and connections with physical attributes (Sims, 2023). Traditional evolutionary algorithms have long navigated this space by applying simplistic, hand-crafted genetic operators, such as adding or deleting nodes and edges (Wang et al., 2019; Sims, 2023; Cheney et al., 2014). However, these operations are semantically blind; they modify the graph's structure without any inherent understanding of the physical consequences, leading to inefficient exploration. More recent learning-based approaches also build upon this graph abstraction, using graph neural networks (GNN) and reinforcement learning (RL) to learn a policy over a set of predefined graph modification actions (Yuan et al., 2022; Dong et al., 2023). While this learns a more intelligent strategy for applying edits, the edits themselves are still constrained to the same limited set of graph operations. Attempts utilizing LLMs for robot design often fall into similar traps. Some rely on predefined grammar rules that restrict the search to specific families (Qiu et al., 2024), while others focus primarily on parameter tuning within fixed skeletal graphs rather than open-ended topological generation (Fang et al., 2025). In essence, current paradigms sidestep the original, more challenging problem of direct XML generation. By operating within a constrained action space, they simplify the problem at the cost of sacrificing the vast expressiveness and higher degrees of freedom inherent to the true design space of XML definitions, thereby limiting their creative potential.

Recent advancements in large language models (LLMs) have demonstrated their remarkable creative capabilities of generating raw forms of structured language (Achiam et al., 2023; Guo et al., 2025), such as computer codes (Chen et al., 2021). In particular, powerful LLMs combined with automated evaluators have proven effective at evolving and optimizing complex algorithms, as highlighted by the groundbreaking AlphaEvolve project (Novikov et al., 2025; Sharma, 2025). This success unlocks a new possibility for robotic design: directly operating on XML definitions in their original, open form. Inspired by this paradigm, we introduce MorphoGen, a novel framework that treats a robot's XML genotype as a form of specialized code and reframes morphology design as a code generation problem. Our core idea is to replace the predefined, hand-crafted genetic operators of traditional evolutional algorithms with an LLM that is prompted to perform semantically rich mutation and crossover operations (Figure 1). To guide this process, we develop a critic LLM that provides gradient-like textual guidance to the primary coding LLM, steering the evolution of robot morphologies. Our approach allows the LLMs to apply their understanding of structure and syntax to directly iterate on raw XML files, enabling an unconstrained and efficient exploration of the design space that lies outside the valid solution spaces of grammar-based methods. To improve the efficiency of evolution, we further propose structure pretraining to serve as high-performing "parent" robots and a two-stage fast controller optimization to accelerate the evaluation of each design. MorphoGen thereby integrates the generative power of LLMs with evolutionary search to design robots in their raw format, unconstrained by the abstractions or limited action spaces of previous methods.

The main contributions of this paper are as follows: (1) We propose an LLM-driven code evolution paradigm for robot design that directly iterates on raw XML definitions of robot morphology. This removes the need for intermediate mediators, thereby addressing the design problem in its original, unconstrained form. (2) Our MorphoGen framework enables efficient morphology optimization through the use of a critic LLM offering gradient-like textual prompts to guide the coding LLM and an accelerated evaluation process for candidate designs. (3) We conduct extensive experiments on challenging locomotion benchmarks, which validate that our approach generates high-performing morphologies that significantly outperform strong baselines by over 52.9% in downstream motoring performance and discovers a diverse range of novel designs.

## 2 RELATED WORKS

### 2.1 ROBOT MORPHOLOGICAL DESIGN

Robot morphological design is a complex bilevel optimization problem (Kim et al., 2021; Hu et al., 2023; He & Ciocarlie, 2024; Jiang et al., 2021; Lu et al., 2025; Cheney et al., 2016; Lu et al., 2025;

Pathak et al., 2019), with the outer level optimizing the structure and the inner level refining the control policy for motion. Traditional approaches such as genetic algorithms (Datta et al., 2015; Brodbeck et al., 2015; Liu et al., 2022) and evolutionary strategies (Xu et al., 2021; Cheney et al., 2018; Zhao et al., 2020), rely on heuristic or search-based methods, which are constrained by predefined search spaces and require explicit design rules. Recent RL-based methods (Yuan et al., 2022; Dong et al., 2023; Singh et al., 2022) have advanced control policy optimization for various skeletons but handle structures indirectly through parameterized representations, often imposing constraints for feasibility. In contrast, our approach directly evolves the robot's skeleton by modifying its underlying XML file, enabling exploration of a broader, unconstrained design space and facilitating the discovery of novel, efficient configurations.

## 2.2 LLM-BASED EVOLUTIONARY FRAMEWORKS

Recent studies (Tian et al., 2025; Liu et al., 2025; Yang et al., 2023) investigate using LLMs as intelligent search operators in evolutionary optimization, leveraging their capacity for code generation and in-context reasoning. For example, RoboMorph (Qiu et al., 2024) represents robot designs as grammar strings and uses a best-shot prompting approach to iteratively propose new rules. RoboMoRe (Fang et al., 2025) focuses on the co-design of morphology and reward functions and utilizes masked templates for parameter optimization. LASeR (Song et al., 2025), incorporate reflection mechanisms to encourage structural diversity. However, these methods fundamentally constrain the search space to a narrow subspace by relying on predefined data structures, and they lack dedicated efficiency optimizations for text-based evolution process. This results in generated morphologies that often exhibit high homogeneity and struggle to surpass the performance of traditional RL-based baselines. To overcomes these challenges, our approach reframes morphology design as unconstrained XML code generation, with a novel critic LLM to provide directional guidance and hierarchical proxy fitness to enhance search efficiency in the complex design space.

## 3 PRELIMINARIES

### 3.1 ROBOT MORPHOLOGY

A robot morphology $M$ is formally defined as a Kinematic Tree (Gupta et al., 2022; Liu et al., 2024) $\mathcal{T} = (\mathcal{V}, \mathcal{E}, r)$, where $\mathcal{V} = \{b_0, b_1, \ldots, b_n\}$ denotes a set of rigid bodies. Each body $b_i$ is associated with a set of physical attributes $\Phi_i$. The element $r \in \mathcal{V}$ is the root body, representing the base of the kinematic hierarchy. $\mathcal{E}$ denotes a set of directed edges where $e_{ij} = (b_i, b_j)$ represents a kinematic connection from the parent body $b_i$ to the child body $b_j$. Each edge $e_{ij}$ is associated with a set of parameters $\Theta_{ij} = (J_{ij}, A_{ij})$, where $J_{ij}$ defines the joint properties and $A_{ij}$ defines the actuation properties that control the joint movement. This Kinematic Tree $\mathcal{T}$ can be translated to text-based XML definition $\mathcal{X}$ using a single-valued serialization function $\mathcal{F}$, such that $\mathcal{X} = \mathcal{F}(\mathcal{T})$.

### 3.2 EVOLUTIONARY OPTIMIZATION

Begein with an initial morphology $\mathcal{X}_0$, the framework proposes candidate modifications to produce successive XML structures $\mathcal{X}_t$ for $t = 1, 2, \ldots, T$, where $T$ is the maximum number of iterations. To evaluate the quality of a robot structure $\mathcal{X}$, a motion controller $\pi_{\mathcal{X}}$ is derived specifically for its morphology. The fitness of $\mathcal{X}$ with $\pi_{\mathcal{X}}$ is assessed in a physics-based simulator, yielding a scalar score $\mathcal{L}(\mathcal{X}, \pi_{\mathcal{X}})$, which quantifies the locomotion efficiency. The robot morphology optimization problem is thus formulated as:

**Input**: An initial XML structure $\mathcal{X}_0$, and an initial motion controller $\pi_0$.

**Output**: An optimized XML structure $\mathcal{X}^*$, along with its associated controller $\pi_{\mathcal{X}^*}$.

**Objective**: Find the morphology $\mathcal{X}^* = \arg\max_{\mathcal{X} \in \Omega_{\mathcal{X}}} \mathcal{L}(\mathcal{X}, \pi_{\mathcal{X}})$ that maximize the fitness.

## 4 METHOD

We propose MorphoGen, an LLM-based evolutionary framework that directly iterates on the robot's raw XML genotype to generate high-performing skeletal structures. As illustrated in Figure 2, Mor-

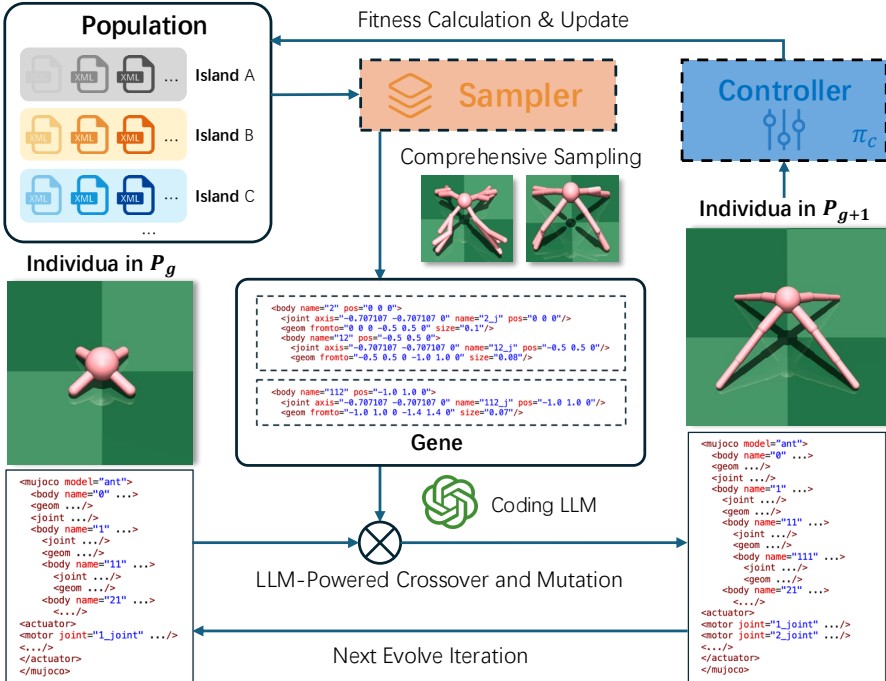

Figure 2: Overview of the MorphoGen evolutionary pipeline.

phoGen employs a guided evolutionary search strategy. Begin with initial robot morphologies, a critic LLM is employed to generating textual feedback that critiques their structural strengths and weaknesses. This feedback serves as a rich, gradient-like textual prompt guiding the coding LLM to perform semantically-aware mutation and crossover operations directly on the parent XML genotypes to produce a new generation of optimized offspring morphologies. Subsequently, each newly generated XML genotype is evaluated with an efficient control policy fine-tuning process to determine its fitness. This closed-loop process, driven by textual critique and semantic code generation, transforms the design search from blind and limited modifications into an intelligent, targeted exploration. Consequently, MorphoGen navigates the vast XML definition space in its original, open form with remarkable efficiency, discovering novel, high-performing robot morphologies that surpass the capabilities of traditional methods. Appendix C provides the prompts utilized within MorphoGen.

## 4.1 LLM DRIVEN EVOLUTIONARY FRAMEWORK

Drawing inspiration from AlphaEvolve (Novikov et al., 2025), the core components of evolution are tailored towards robotic design with redefined XML context as follows:

- **Individua** $M$ is a complete XML definition, which serves as the robot's genotype by explicitly defining its entire skeletal structure and joints, and physical properties. This genotype is the direct optimization target of our framework.

- **Genes** $\{G_1, G_2, \ldots G_k\} \in M$ are conceptualized as specific and semantically meaningful blocks within the XML definition, which are the core elements manipulated during evolution.

- **Island** $I = \{M_1, M_2, \ldots, M_n\}$ is a collection of $n$ such individuals.

- **Population** $P = \{I_1, I_2, \ldots, I_m\}$ is a collection of $m$ islands, which contains all the individuals.

Designing robot morphology differs drastically from code generation of AlphaEvolve (Novikov et al., 2025) in every stage of the pipeline, and we propose parent selection, crossover and mutation, as well as fitness calculation, that prioritize morphology understanding, XML genotype modification, and locomotion optimization. In specific, one generation $P_g$ of robot morphology XML definitions are evolved to the next $P_{g+1}$ as follows:

**Comprehensive Sampling.** To form the parent pool for the subsequent generation from the current population $P_g$, we employ a hybrid selection strategy designed to balance the exploitation and exploration through a combination of three criteria: (1) Elitism (Qiu et al., 2024): The top individuals

Figure 3: Illustration of the efficient evolutionary optimization process in MorphoGen.

with the highest fitness scores of each island are selected to preserve and propagate high-performing genotypes. (2) Diversity (Schulman et al., 2017; Mouret & Clune, 2015): To avoid premature convergence to local optima and promote exploration of diverse regions in the design space, individuals structurally dissimilar to others that maximize Tree Edit Distance $TED(M_i, M_j)$ on the XML structure are chosen. (3) Randomness (Fang et al., 2025): A small subset of individuals is chosen uniformly at random to maintain a reservoir of potentially valuable XML blocks not present in high-fitness or diverse candidates. This multi-pronged selection strategy ensures that the evolutionary search is simultaneously greedy, exploratory, and robust.

**LLM-Powered Crossover and Mutation.** Unlike traditional evolutionary algorithms that rely on simplistic, random operators, MorphoGen leverages a coding LLM to perform semantically coherent XML modifications. In the crossover operation, an offspring $M_{\text{child}}$ is generated by intelligently combining structural features from several parent individuals, $\{M_{p_1}, M_{p_2}, \ldots, M_{p_i}\}$. The LLM is prompted to merge high-quality XML blocks from all parents, producing a structurally sound offspring that inherits meaningful traits. In the mutation operation, novel variations are introduced into a single parent $M_p$, modifying its XML genotype to reach better motion performance.

**Fitness Calculation.** After generating a new XML genotype $M$, we fix its structure and employ RL algorithm (Yuan et al., 2022) to develop an adaptive controller $\pi_C$ for locomotion. The RL training process optimizes $\pi_C$ over a fixed number of steps in a simulated environment, after which the control policy is deployed to evaluate the robot's performance. The fitness $F$ of the skeletal structure $M$ is quantified by measuring the Euclidean distance traveled from the starting point within a fixed time horizon in the simulation. This evaluation provides a robust indicator of the structure's effectiveness, leading to guided population update for the next generation. Details of robot controller optimization is provided in Appendix B.

## 4.2 EFFICIENT EVOLUTIONARY OPTIMIZATION

Evolutionary algorithms are notoriously sample-inefficient, which is even worse in robot design due to the costly inner-loop optimization required to evaluate each skeletal structures. MorphoGen addresses this bottleneck with three key innovations: structure pretraining to initialize the population with high-quality genotypes, text-based gradient guidance to enable targeted optimization, and hierarchical proxy fitness to accelerate evaluation. Together, these components transform the evolutionary search into a more efficient and directed process.

**Structure Pretraining.** Traditional robot morphology design heavily relies on expert knowledge, and initializing an LLM to explore the expansive structure space from scratch is computationally prohibitive. To overcome this, MorphoGen leverages domain expertise by initializing the population $P_0$ with a diverse set of high-quality XML genotypes derived from expert-designed morphologies or existing advanced methods. These pre-trained skeletal structures encode rich domain knowledge about effective robot postures and dynamics. Benefiting from this informed gene pool, the coding LLM focuses its exploration on high-potential regions of the design space, significantly enhancing the quality of generated skeletal structures and the efficiency of the evolutionary search.

**Text-based Gradient Guidance.** Unlike traditional evolutionary algorithms, where random mutations often lead to degraded performance, MorphoGen employs a critic LLM to provide targeted

optimization guidance. For each XML definition, the critic LLM analyzes the skeletal structure and generates textual feedback that identifies the likely structural causes for the motion performance, highlighting both structural strengths and weaknesses. This guidance is then integrated into the prompt for the coding LLM, guiding it to perform semantically-aware mutations that preserve advantageous features while addressing identified deficiencies. This guided approach ensures that each evolutionary step is more likely to be a meaningful improvement, dramatically increasing the directional efficiency of the search.

**Proxy Fitness.** Learning an optimal control policy for each XML genotype $M_i$ from scratch is computationally expensive, often requiring millions of simulation interactions. However, recognizing that an offspring is often a small modification of its parent, we posit that their optimal control policies are also close in the parameter space, which enables efficient policy reuse. Therefore, instead of full retraining, we can approximate the best control policy of current morphology by briefly fine-tuning the parent's optimized policy. Moreover, with potentially thousands of generated skeletal structures throughout the whole process, even fine-tuning can be time-consuming. To overcome the above challenges, MorphoGen first employs a basic controller $\pi_0$ which is pre-trained for initial robot morphology, and then utilizes a two-stage hierarchical proxy fitness evaluation. Specifically, in the first stage, each skeletal structures $M_i$ are directly controlled with $\pi_0$ and we can simply consider the corresponding locomotion scores as the fitness $F_i^1$. This evaluation is extremely fast and does not require any additional training costs. We take $F_i^1$ as an initial screening criterion to ignore those structures that have obvious functional issues. For the skeletal structures with $F_i^1$ greater than a certain threshold, which indicates a promising candidate, the second stage is triggered. In the second stage, $\pi_0$ will be fine-tuned against structure $M_i$ for several steps to get an adaptive controller $\pi_i$ which is applied to calculate the locomotion score as fitness $F_i^2$. The final proxy fitness $F_i$ is thus the result of this conditional, two-stage process. This hierarchical approach drastically reduces the average evaluation time per individual, allowing MorphoGen to assess a much larger volume of candidates and conduct a more thorough and effective exploration of the design space.

In summary, through an integrated framework of LLM-driven evolution, gradient-like textual guidance, and accelerated fitness evaluation, we achieve direct optimization in the raw XML definition space with improved efficiency. We now validate the effectiveness of this approach experimentally.

## 5 EXPERIMENTS

### 5.1 EXPERIMENT SETTINGS

**Baselines** We compare MorphoGen with the following baselines: Neural Graph Evolution (NGE) (Wang et al., 2019), Evolutionary Structure Search (ESS) (Cheney et al., 2018), Random Graph Search (RGS) (Wang et al., 2019), Transform2Act (T2A) (Yuan et al., 2022), Symmetry-Aware Robot Design (SARD) (Dong et al., 2023), RoboMorph (Qiu et al., 2024) and RoboMoRe (Fang et al., 2025). For the LLM-based methods, we adopt Qwen3 (Yang et al., 2025) as the backbone model. More details are provided in Appendix A.

**Evaluation** Evaluations are conducted in the MuJoCo simulator across four environments: 2D Locomotion (Hopper), 3D Locomotion (Ant), Swimmer, and Gap Crosser (Gap) (Todorov et al., 2012). Each robot is controlled by a controller learned via an RL method (Yuan et al., 2022), and performance is measured by the distance traveled within a fixed time. Specifically, NGE, ESS, RGS, and our method learn control policies for fixed skeletal structures with PPO (Schulman et al., 2017). In contrast, RL-based methods like T2A and SARD optimize the skeletal structure and control policy simultaneously. To ensure a fair comparison with these baselines, MorphoGen uses reference results from only the first half of T2A's optimization steps, rather than its fully optimized solutions. In addition to evaluating motion efficiency with an optimal controller, we also wonder how quickly a functional controller can be obtained with limited time and computational resources. To this end, we evaluate each structure using two distinct controllers: a fine-tuned controller (FC), trained for only 0.3M steps, and a fully trained controller (FTC), which is trained until convergence.

### 5.2 OVERALL PERFORMANCE

To completely demonstrate the quality and diversity of the generated skeleton structures of our methods, we report the top three results, and with name MorphoGen$^i$ denotes the top $i$-th result. We

| Method | Fine-tuned Controller (FC) | | | | Fully Trained Controller (FTC) | | | |
|---|---|---|---|---|---|---|---|---|
| | **Hopper** | **Ant** | **Swimmer** | **Gap** | **Hopper** | **Ant** | **Swimmer** | **Gap** |
| ESS | $163.4_{\pm14.1}$ | $108.9_{\pm10.1}$ | $93.1_{\pm11.5}$ | $99.6_{\pm13.0}$ | $658.7_{\pm52.7}$ | $1387.5_{\pm113.3}$ | $349.0_{\pm28.9}$ | $218.6_{\pm19.7}$ |
| RGS | $483.5_{\pm54.4}$ | $508.3_{\pm31.0}$ | $107.8_{\pm13.8}$ | $240.7_{\pm26.3}$ | $849.5_{\pm69.0}$ | $1394.2_{\pm79.0}$ | $301.4_{\pm31.5}$ | $670.8_{\pm39.8}$ |
| NGE | $1725.9_{\pm115.3}$ | $896.2_{\pm41.8}$ | $119.0_{\pm7.0}$ | $601.4_{\pm40.6}$ | $2647.1_{\pm224.0}$ | $2015.3_{\pm205.4}$ | $513.9_{\pm42.2}$ | $822.3_{\pm58.0}$ |
| T2A | $2143.2_{\pm192.2}$ | $1140.2_{\pm136.2}$ | $703.4_{\pm24.6}$ | $651.0_{\pm76.3}$ | $\mathbf{8130.4}_{\pm1093.8}$ | $4268.7_{\pm351.4}$ | $877.4_{\pm65.9}$ | $2147.5_{\pm322.6}$ |
| SARD | $1841.4_{\pm210.4}$ | $1174.8_{\pm79.6}$ | $970.7_{\pm17.3}$ | $896.3_{\pm67.2}$ | $7109.0_{\pm839.3}$ | $\mathit{4347.1}_{\pm391.7}$ | $892.8_{\pm66.6}$ | $2532.7_{\pm366.2}$ |
| RoboMorph | / | $853.3_{\pm60.8}$ | / | / | / | $2842.3_{\pm353.4}$ | / | / |
| RoboMoRe | $1612.0_{\pm193.2}$ | $1022.4_{\pm87.8}$ | $808.1_{\pm17.2}$ | $718.8_{\pm68.8}$ | $6697.1_{\pm492.4}$ | $3514.1_{\pm278.9}$ | $693.9_{\pm34.5}$ | $2833.8_{\pm286.5}$ |
| MorphoGen[1] | $\mathbf{2474.1}_{\pm163.6}$ | $\mathbf{1560.4}_{\pm110.3}$ | $\mathbf{1056.3}_{\pm15.5}$ | $\mathbf{969.1}_{\pm62.4}$ | $\underline{7409.0}_{\pm654.2}$ | $\mathbf{4740.6}_{\pm282.6}$ | $\mathbf{1058.0}_{\pm37.4}$ | $\mathbf{5652.7}_{\pm280.5}$ |
| MorphoGen[2] | $2265.2_{\pm135.8}$ | $1425.5_{\pm71.3}$ | $1024.4_{\pm15.1}$ | $946.7_{\pm66.0}$ | $7127.9_{\pm741.5}$ | $\underline{4357.9}_{\pm267.2}$ | $\underline{1029.4}_{\pm45.2}$ | $\underline{5258.3}_{\pm193.8}$ |
| MorphoGen[3] | $\mathit{2215.0}_{\pm123.5}$ | $\mathit{1181.3}_{\pm74.3}$ | $\mathit{1008.6}_{\pm19.5}$ | $860.8_{\pm57.7}$ | $6384.4_{\pm307.2}$ | $3994.2_{\pm233.4}$ | $\mathit{960.1}_{\pm17.2}$ | $\mathit{3877.7}_{\pm141.0}$ |

Table 1: Comparison of locomotion score of the generated skeleton structures across different methods, environments and controller. The **bold** values indicate the best-performing solutions, the underlined values denote the second-best solutions, and the *italicized* values represent the third-best solutions (determined separately per column). Each robotic skeleton structure is evaluated using control policies trained across six seeds.

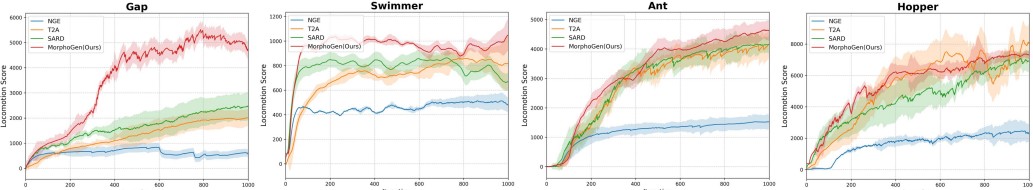

Figure 4: Training curves of locomotion metric, comparing MorphoGen with baseline methods.

summarize the motion performance of the generated robot skeleton structure in Table 1, from which we can have the following observations,

- **Traditional search or evolutionary algorithms fail to provide quality skeleton structures.** The solution space of robot skeletons is extremely large and highly non-convex. Without a clear exploration direction or strong priors, methods such as NGE, ESS, and RGS tend to converge slowly and often produce degenerate structures with poor locomotion ability. These approaches consistently yield the lowest performance, with an average degradation of 76.2% compared to RL-based methods, highlighting the necessity of providing optimizing directions.

- **Constraints on the design space severely limit the quality of generated solutions.** Leading approaches such as T2A, RoboMorph, and RoboMoRe fundamentally restrict the search process by relying on predefined grammar rules, fixed graph topologies, or limited parameter tuning within static skeletons. While these abstractions simplify the optimization problem, they confine the evolutionary search to a narrow, pre-specified subspace, preventing the exploration of the full morphological manifold. As a result, these methods struggle to escape local optima, yielding morphologies with limited locomotion capabilities. Empirically, the average performance of morphologies generated by these constrained baselines is only 63.5% of that achieved by MorphoGen. In contrast, our method operates directly on the raw XML genotype without prior structural assumptions or fixed action spaces. This unconstrained flexibility allows MorphoGen to navigate the immense combinatorial space effectively, unlocking novel, high-performing designs that are inaccessible to methods bound by rigid design priors.

- **Our approach has significant advantages over other methods.** Our method consistently generates skeleton structures that achieve superior locomotion performance across diverse environments. Evaluated by the FTC, our approach attains the best results in the majority of skeletons and ranks within the top two in all cases, yielding an average improvement of 57.5% over state-of-the-art RL baselines. Notably, our method still outperforms all baselines by at least 29.5%, and dominates the top three results in every environment with the FC. These findings indicate that our approach not only enables diverse and high-quality skeleton generation, but also demonstrates strong potential for real-world applications where training resources are constrained.

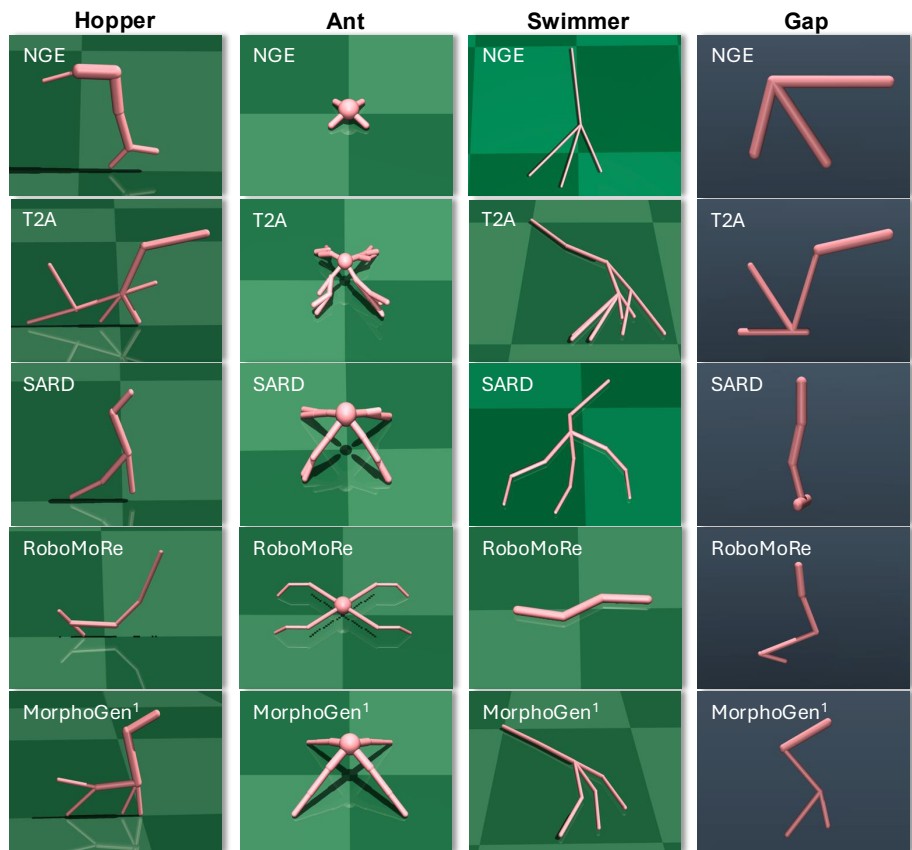

Figure 5: Visualization of representative robot morphologies generated by different methods.

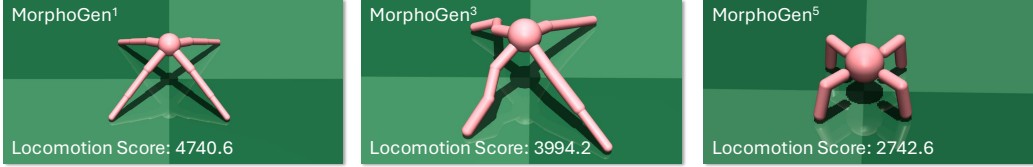

Figure 6: Top-performing morphologies generated by MorphoGen in the Ant environment.

To further showcase the efficacy of our approach, we visualize the motion performance of skeleton structures generated by different methods throughout the training process. As shown in Figure 4, our method significantly outperforms baseline approaches with the FTC, achieving an average performance improvement of 40.5% across all environments. Moreover, our approach generates skeleton structures that facilitate rapid learning of effective control policies. Specifically, while baseline methods often require hundreds of optimization iterations to achieve acceptable locomotion performance, our method consistently produces skeletons that achieve to effective motion strategies in fewer than 100 iterations.

## 5.3 STRUCTURAL ANALYSIS

We provide visual images of skeletal structures generated by different methods, as shown in Figure 5. In contrast to the intricate and often incomprehensible structures generated by RL-based methods, the skeletons produced by our approach are clear, intuitive, and have greater potential for real-world implementation. These designs achieve superior motion performance with fewer movable joints, indicating a higher level of locomotion efficiency. Our method also naturally evolves various bio-inspired structures that resemble real-world organisms. For example, the generated ant skeleton resembles a spider, the hopper skeleton looks like a horse, and the swimmer skeleton mim-

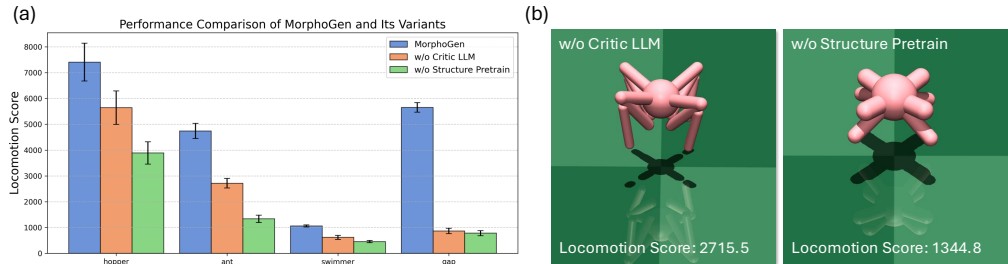

Figure 7: Ablation study on key components of MorphoGen. (a) The performance comparison between MorphoGen and its variants. (b) The corresponding generated robot morphologies.

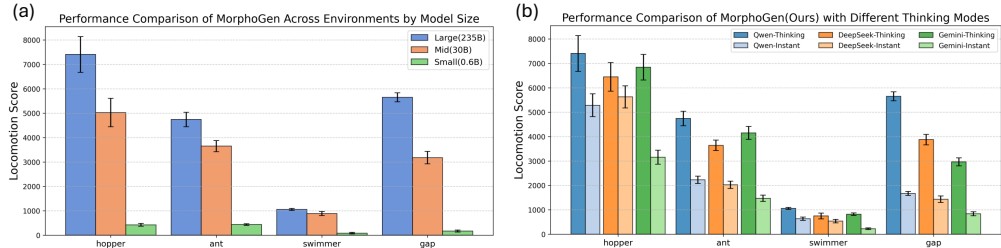

Figure 8: Backbone model selection analysis. (a) The performance of Qwen3 with varying parameter quantities as the backbone model. (b) Performance comparison of models with and without a reasoning component.

ics an octopus. This demonstrates our method's ability to discover effective and elegant designs by leveraging principles found in nature, which are possibly sourced from the strong priors in LLMs. Furthermore, our approach is capable of generating a diverse range of high-performance skeletons. Figure 6 illustrates the top-performing ant skeletons generated by MorphoGen which achieve high scores while possessing distinct structures and locomotion gaits (see Appendix E for more details). This highlights our method's capacity to explore a wide variety of effective design solutions rather than converging on a single, narrow set of configurations.

### 5.4 ABLATION STUDY

To investigate the effectiveness of the employed critic agent and the initial elite solutions, we separately remove these two key components from our framework and report the results of variants in Figure 7. The findings indicate that removing either component significantly degrades the performance of the generated robot skeletons, with the average motion score of the generated structures decreasing by 50.5% and 65.7% respectively. Specifically, the absence of feedback from the critic agent deprives the LLM of a targeted optimization direction during the evolutionary process, severely hindering its exploration efficiency. This is particularly evident in the design of the gap crosser, where the motion score plummets from 5652.7 to 866.4, a significant drop of 84.7%. Meanwhile, without the initial elite solutions, the LLM lacks a crucial set of high-quality examples to learn from. This forces it to waste valuable computational resources on low-quality, sub-optimal designs, as it fails to acquire essential domain knowledge. A prime example is the ant structure, where the motion efficiency of the generated skeletons is only 28.2% of what our full framework achieves. These results underscore the critical role of both the critic agent and the initial elite solutions in enabling our LLM-based framework to generate high-performing skeletal structures.

### 5.5 BACKBONE MODEL SELECTION

To investigate whether different backbone models influence skeleton design, we conducted experiments focusing on the impact of model size and whether a model is specifically optimized for reasoning. As presented in Figure 8, the size of the backbone model has a significant effect on the performance of the generated skeleton structures. As the model's parameter scale decreases, the locomotion scores of the produced skeletons drop sharply, from 7409.0 to 424.3(-94.3%), aligning

with established scaling laws in LLMs. Furthermore, models specifically designed for reasoning significantly boost the framework's ability to generate higher-quality motion skeletons. With their enhanced logical and causal reasoning capabilities, are better equipped to understand the intricate relationships between a skeleton's structure and its physical dynamics. It allows the LLM to more effectively deduce which structural modifications are likely to lead to performance improvements, resulting in more intelligent and efficient evolutionary steps.

## 6 DISCUSSION

In this work, we introduced MorphoGen, an evolutionary framework that reframes the automated design of robot morphologies as a code generation problem. By leveraging LLMs to directly evolve the raw XML genotypes of robots, our approach circumvents the need for intermediate graph-based abstractions and predefined modification actions that seriously constrain the exploration space. Through a combination of a critic LLM providing gradient-like textual guidance , an efficient hierarchical fitness evaluation , and an initial population of high-quality structures , MorphoGen demonstrates a remarkable ability to navigate the vast and complex design space. Our experiments show that MorphoGen not only discovers diverse and high-performing morphologies, but also significantly outperforms existing baselines across a suite of challenging locomotion tasks.

Looking ahead, a promising direction is to enhance the framework's creative potential by fostering collaboration among multiple, specialized LLMs. Moreover, achieving a co-evolution of morphology and control policy is also critical. By leveraging the LLM's understanding of the link between form and function, it could suggest control parameters tailored to the specific morphology. This would create a tighter integration between the evolution of the "body" and the "brain," potentially accelerating the evaluation bottleneck and leading to more holistic and efficient robotic systems.

## ETHICS STATEMENT

This work focuses on the automated design of robot morphologies using LLMs in simulated environments. All evaluations are conducted in physics simulators, ensuring no direct risks to safety, privacy, or the environment. No conflicts of interest or external sponsorships apply, and all methods comply with research integrity standards. We confirm adherence to the ICLR Code of Ethics.

## REPRODUCIBILITY STATEMENT

We release the complete source code for the MorphoGen framework anonymously, including all scripts required to replicate our experiments. The release will include the set of expert-designed XML files used for structure pretraining, the specific prompts for the coding and critic LLMs, and all hyper-parameters for both the evolutionary search and the control policy optimization.

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

## A  BASELINE DETAILS

We provide additional implementation details for all baselines below.

- **Neural Graph Evolution (NGE)** (Wang et al., 2019) formulates robot design as a graph search problem and employs graph neural networks as controllers to share policies across different morphologies. This significantly reduces evaluation cost during search and enables efficient discovery of kinematically meaningful designs.

- **Evolutionary Structure Search (ESS)** (Cheney et al., 2018) jointly evolves morphology and control in soft robots. It introduces *morphological innovation protection*, which temporarily relaxes selection pressure on recently mutated morphologies, allowing controllers to readapt and thus preventing premature convergence.

- **Random Graph Search** (Wang et al., 2019) performs naive random sampling of graph-structured morphologies without guided exploration. Although simple, it serves as a strong baseline to highlight the advantage of structured evolutionary search.

- **Transform2Act** (Yuan et al., 2022) treats robot design as a conditional decision-making process and learns policies that jointly determine both the morphology and control. By leveraging reinforcement learning with modular architectures, it achieves improved sample efficiency over pure evolutionary approaches.

- **Symmetry-Aware Robot Design (SARD)** (Dong et al., 2023) incorporates group-theoretic symmetry constraints into the robot design process. By searching for and enforcing optimal symmetry subgroups, it reduces the design space, improves control stability, and produces more efficient morphologies across diverse tasks.

- **RoboMorph** (Qiu et al., 2024) employs a structured grammar library to describe the sequence of morphology designs. An evolutionary loop retains the highest-fitness individuals as parents for the next generation.

- **RoboMoRe** (Fang et al., 2025) fixes the graph structure of morphology and searches for the best parameters of limbs. It follows a coarse-to-fine paradigm and jointly optimizes morphology and the reward function.

## B  CONTROLLER OPTIMIZATION

The controller optimization for each generated robot structure is performed using RL to develop an adaptive control policy $\pi_C$ for locomotion tasks. The problem is formulated as an infinite-horizon discounted Markov Decision Process (MDP), defined by the tuple $\mathcal{M} = (\mathcal{S}, \mathcal{A}, P, r, \rho_0, \gamma)$, where $\mathcal{S}$ is the state space, $\mathcal{A}$ is the action space, $P : \mathcal{S} \times \mathcal{A} \to \mathcal{S}$ is the transition probability function, $r : \mathcal{S} \times \mathcal{A} \to \mathbb{R}$ is the reward function, $\rho_0 : \mathcal{S} \to [0, 1]$ is the initial state distribution, and $\gamma \in [0, 1)$ is the discount factor. For a fixed robot morphology, the objective is to learn a control policy $\pi_C : \mathcal{S} \to \mathcal{A})$ that maximizes the expected cumulative discounted reward:

$$J(\pi_C) = \mathbb{E}_{s_0 \sim \rho_0, a_t \sim \pi_C(\cdot|s_t), s_{t+1} \sim P(\cdot|s_t, a_t)} \left[ \sum_{t=0}^{\infty} \gamma^t r(s_t, a_t) \right],$$

where $s_t \in \mathcal{S}$ and $a_t \in \mathcal{A}$ are the state and action at time $t$.

We employ Proximal Policy Optimization (PPO) (Schulman et al., 2017) to optimize the policy, following the approach in Yuan et al. (2022) for locomotion control.

## C  PROMPTS

**Coding LLM**

System Message:

You are an expert in robotics, MuJoCo simulation, and robot morphology design. Your task is to optimize a robot design defined in MuJoCo XML format to maximize locomotion performance while adhering to best practices.

Key Design Principles:
1. Structural Stability: Ensure a robust base and secure joint connections for balance.
2. Locomotion Efficiency: Design limbs and joints to maximize forward movement efficiency.
3. Actuator Coverage: Assign motor actuators to all movable joints.
4. Physical Realism: Use realistic sizes, masses, and joint ranges for simulation accuracy.
5. Complexity Balance: Optimize capability without overly complex structures that hinder control.

XML Structure Requirements:
- Position the root body at an appropriate height for stability.
- Ensure all hinge joints have corresponding motor actuators in the ¡actuator¿ section.
- Match joint names exactly between ¡joint¿ and ¡motor¿ elements.
- Define realistic joint ranges for natural movement.
- Specify capsule geometries with accurate fromto coordinates.
- Maintain valid MuJoCo XML syntax and structure.

Please modify one aspect of the robot design per iteration to isolate improvements: limbs, joints, or bodies.
- Adjust limb lengths and orientations to enhance locomotion efficiency.
- Strategically add or remove body segments to balance capability and simplicity.
- Optimize body positions and orientations for improved stability and movement. - Ensure design complexity remains controllable.

Output:
- Do not include explanations, comments, or additional text outside the XML.
- Return only the complete, valid MuJoCo XML."

User Message:
Current Morphology Information:
- Morphology: {current_xml}
- Locomotion Score: {current_fitness}
- Structure Tree: {current_structure_tree}
- Focus areas: {current_critic_ouput}

Sampled Morphology Information:
- Morphology: {sampled_xml}
- Locomotion Score: {sampled_fitness}
- Structure Tree: {sampled_structure_tree}
- Focus areas: {sampled_critic_ouput}

Program Evolution History: {evolution_history}

Task: Rewrite the program to improve its Locomotion Score. Provide the complete new xml code.

# Your modified xml here

**Critic LLM**

[t] System Message:

> You are a robotics expert acting as a critic for an evolutionary algorithm designing a Mu-JoCo robot. Your task is to provide a specific, actionable suggestion for how to improve the robot's design. Do not give general or too specific advice. Your suggestion should be a concrete change to the XML structure related to an aspect.

User Message:

> The robot's current XML structure is: xml {current program} Provide a concise suggestion for improvement focusing specifically on one aspect from {aspects}. Example for limb lengths and orientations: Try increasing the length of the front leg segments. Example for body segments: Consider adding a new body segment or deleting an existing body segment to the torso for more flexibility. Example for body positions and orientations: Adjust the initial orientation of the main torso to be more parallel to the ground. Also, please briefly point out the weakness and strengths in the current structure.

> Your suggestion:

## D   SENSITIVITY ANALYSIS

Utilizing strong initial genotypes is a widely adopted practice (Qiu et al., 2024; Fang et al., 2025; Yuan et al., 2022) to ensure the search starts within a functionally meaningful region, thereby maximizing optimization efficiency. However, the initial population may cause profound impact to the evolution process and lead to totally different solutions. To investigate to what degree the initial genotypes will influence the final morphology, we conduct an ablation study that use different initial genotypes. Specifically, We employ the morphologies generated by T2A (Yuan et al., 2022) (which is asymmetry with multi-joint and long limbs) and NGE (Wang et al., 2019) (which is symmetric with single-joint and short limbs) and try each possible combinations. Moreover, we also add a 10% random perturbation (rp) to the structure parameters. to showcase the robustness. We summarize both the characteristics and the locomotion scores of the generated morphologies in Table 2.

| Variant | T2A | T2A(rp) | NGE | NGE(rp) | T2A+NGE | T2A(rp)+NGE(rp) |
|---|---|---|---|---|---|---|
| Characteristics | AS
MJ
LL | AS
MJ
LL | S
SJ
SL | S
SJ
SL | S
MJ
LL | S
MJ
LL |
| Locomotion Score | 3858.7 | 3764.2 | 2478.5 | 2608.9 | 4740.6 | 4415.2 |

Table 2: Ablation study on different initial genotypes, where "rp" denotes random perturbation, "AS" denotes to asymmetry, "S" denotes to symmetric, "MJ" denotes to multi-joint, "SJ" denotes to single-joint, "LL" denotes to long limbs and "MJ" denotes to short limbs.

While the structural characteristics influence the final evolutionary direction, experiments with random perturbation show minimal performance drops. This confirms that the search process is robust and effective as long as the initial population contains a diverse set of potentially beneficial locomotion structures. Moreover, the similar characteristics between the genotypes and final solutions confirm that the final evolved designs are not replicas of the initial expert structures. The initial seeds serve only as high-level guidance.

To quantitatively analyze this deviation, we compute the Tree Edit Distance (TED) (Pawlik & Augsten, 2016)-based similarity between the best-evolved solutions and their initial expert genotypes, as domenstrated in Table 3.

The low similarity scores indicate that the generated morphologies are substantially different. They retain only beneficial high-level patterns while evolving highly optimized structural details, validating that MorphoGen performs effective search in the unconstrained XML space.

| Environment | Hopper | Ant | Swimmer | Gap |
|---|---|---|---|---|
| **Similarity** | 54.1% | 22.6% | 43.7% | 35.1% |

Table 3: TED-based structural similarity between the best MorphoGen-evolved solutions and the initial genotypes. Lower values indicate more substantial topological innovation.

## E  DIVERSITY

To rigorously assess the exploration capability of MorphoGen compared to baselines, we employ two distinct metrics: **TED** (Tree Edit Distance) (Pawlik & Augsten, 2016) and **JSD** (Jensen-Shannon Divergence) (Lin, 2002) of length distribution. We measure the pairwise diversity among the top-5 performing morphologies generated by each method in the Ant environment. Specifically, for TED which measures differences in the graph topology, we calculate the pairwise TED between the Kinematic Trees of the morphologies and then normalize and compute the average. For JDS which indicates greater diversity in the parameter space, we first bin the limb lengths of each morphology then calculate the pairwise JSD values of the resulting length distributions and take the average.

| Metric | T2A | RoboMoRe | MorphoGen |
|---|---|---|---|
| TED (Structural Diversity) | 0.183 | 0 | **0.396** |
| JSD (Parameter Diversity) | 0.129 | 0.165 | **0.248** |

Table 4: Morphological and Parameter Diversity. Higher values indicate greater diversity.

The results in Table 4 demonstrate that MorphoGen significantly outperforms existing baseline methods. By operating directly on unconstrained XML and utilizing our explicit selection criteria , MorphoGen achieves superior diversity in both morphological graph structure and structural parameter distributions.

## F  PROXY FITNESS

The primary bottleneck in evolutionary morphology optimization is the extremely high computational cost of training an optimal controller for every candidate design. Our hierarchical proxy fitness strategy is designed to address this challenge by significantly reducing the evaluation time per morphology, thus enabling greater exploration. The time difference between evaluation methods is substantial: a typical proxy fitness calculation spends approximately 2 minutes, whereas training an optimal controller from scratch costs about 3 hours (a $100\times$ increase). Using the full controller for fitness calculation is computationally impractical for extensive evolutionary search.

To study the impact of the proxy fitness on overall optimization performance, we compare three variants on a fixed computational budget, reducing the evolutionary iterations for the full-controller variant to make the study feasible.

| Variant | Iterations | Total Time | Hopper | Ant | Swimmer |
|---|---|---|---|---|---|
| w/o Proxy Fitness (Optimal Controller) | 40 | $\sim$ 108h | 5796.0 | 3862.8 | 789.7 |
| w/ Proxy Fitness (Limited Iterations) | 40 | $\sim$ 1.2h | 5018.3 | 3194.6 | 754.8 |
| w/ Proxy Fitness (Full Search Budget) | 100 | $\sim$ 4h | **7409.0** | **4740.6** | **1058.0** |

Table 5: Performance Comparison of Proxy Fitness Variants.

While using an optimal controller yields slightly higher quality designs when the number of iterations is artificially fixed at 40, its massive computational overhead makes extensive exploration impossible. In contrast, the proxy fitness allows the algorithm to evaluate orders of magnitude more morphologies within the same time frame. By utilizing the proxy fitness and facilitating a larger number of evolutionary steps, MorphoGen achieves 28.2% better overall performance, while consuming 96% fewer compute resources per iteration. This confirms that the efficiency provided by

the hierarchical proxy fitness is essential for effective evolutionary search in the unconstrained XML design space.

## G    USE OF LLMS

In addition to their essential role in generating robot morphologies within MorphoGen, LLMs are used only for polishing some paragraphs. The research ideas and designs are entirely conceived by the authors.

