# OpenReview forum: "MorphoGen: Evolving Robot Morphologies with Large Language Models"
_ICLR.cc/2026/Conference — Submitted to ICLR 2026_

### Official Review · Reviewer_u5DK · 2025-10-27

**Soundness:** 3
**Presentation:** 4
**Contribution:** 2
**Rating:** 6
**Confidence:** 4

**Summary:**

This paper addresses a fundamental challenge in evolutionary Robotics, i.e., robot design automation. An LLM-based evolutionary strategy is proposed, which employs a critic LLM for textual guidance that facilitates more informative and comprehensive evolutionary search. A couple of other techniques, including structure pretraining, two-stage proxy fitness evaluation, and a hybrid sampling strategy, are also proposed that yield additional performance gains. Extensive experiments on simulated locomotion tasks validate the superiority of the proposed method over traditional evolutionary algorithms and RL-based design algorithms.

**Strengths:**

1.The paper is well written and easy to follow. The abundant illustrations and qualitative results greatly help with the reader’s understanding. The analysis in Section 5.3 is particularly interesting, revealing the correspondence between evolved robots and natural priors.

2.The paper tackles a long-lasting challenge in Robotics, and proposes an effective approach that leverages the semantic priors of large language models to aid more efficient search. The algorithmic designs are intuitive and straightforward, yet proved to be competitive experimentally.

3.The authors explicitly consider low-compute scenarios where only a minor fine-tuning budget is allowed, and confirm that the proposed approach is still competitive, adding to its practicality.

**Weaknesses:**

1.The baseline algorithms seem dated. As numerous LLM-based evolutionary strategies have been proposed in recent years (with some of them listed below), the authors are suggested to compare against one or two of them, or at least clarify the relationships with them. The absence of such analysis leaves the paper’s novelty largely unclear.

- Qiu K, Pałucki W, Ciebiera K, et al. Robomorph: Evolving robot morphology using large language models[J]. arXiv preprint arXiv:2407.08626, 2024.

- Song J, Yang Y, Xiao H, et al. Laser: Towards diversified and generalizable robot design with large language models[C]//The Thirteenth International Conference on Learning Representations. 2025.

- Yang C, Wang X, Lu Y, et al. Large language models as optimizers[C]//The Twelfth International Conference on Learning Representations. 2023.

2.The comprehensive sampling strategy introduced in Section 4.1 seem to be a key component, and the increase of morphological diversity is claimed throughout the paper. However, the paper lacks any quantitative measurement and comparison of diversity, with only a couple of examples given in Figure 6.

3.As structure pretraining turns out critical in ablation studies, it becomes questionable regarding how much the performance gains over baseline algorithms are merely due to the additional expert knowledge rather than the algorithm itself.

**Questions:**

1.Beside the fitness of evolved robots, could the authors describe the influence of structure pretraining on the evolutionary dynamics? More specifically, to what extent does the algorithm stick to the structures given in the initial population, and how much does it discover new, beneficial structures?

2.Does the critic LLM also receive fitness scores as input? If not, could the inclusion of such information help the critic LLM better identify beneficial substructures against harmful or neutral ones?

---

> ### Author Response · Authors · 2025-11-21
> **Q1-Q2**
>
> **Q1:** Baseline are dated.
>
> **A1:** Thanks for your critical suggestions. We add two related baselines RoboMorph[1] and RoboMoRe[2] which also incorporate LLMs and evolutionary framework for robot morphology design, and provide a deeper discuss about LLM-based evolutionary algorithm.
>
> **First,** MorphoGen differs fundamentally in its **(1)search space, (2)evolutionary mechanism and  (3)evaluation efficiency.** Below, we clarify the critical technical differences that make MorphoGen a novel and distinct contribution.
>
> - **Search Space.** RoboMorph and RoboMoRe operate within narrow, constrained morphology spaces $\mathcal{M}^{'} \subset \mathcal{M}$. Specifically, RoboMorph does not generate robot files directly. Instead, it generates grammar strings based on the RoboGrammar rule set, limiting the search to a small family of crawler-style bodies. RoboMoRe assumes a fixed skeletal graph and its morphology search focuses heavily on parameter tuning within existing structures rather than open-ended topological generation. In contrast, MorphoGen treats robot design as a Code Generation problem and interacts directly with the raw XML. Noteably, In the 3D Locomotion (Ant) task, the optimal morphology discovered by MorphoGen (see Fig. 5) completely lies outside the valid solution spaces of these baselines.
> - **Evolutionary Mechanism.** RoboMorph and RoboMoRe relies on best-Shot prompting, lacking an explicit feedback signal explaining why a design failed or succeeded. In MorphoGen, however, we introduce a Critic LLM that analyzes the morphology and generates explicit feedback. This feedback serves as a text-based gradient, explicitly guiding the Coding LLM on how to mutate the genotype. This transforms the process from examples-guided random exploration into a directed search.
> - **Evaluation Efficiency.** RoboMorph and RoboMoRe trains controller from scratch for every generated morphology, which waste a lot of resources on evaluation instead of optimization. In contrast, MorphoGen introduces a hierarchical proxy fitness strategy. This allows us to evaluate a significantly larger population of diverse structures with the same computational budget, which is critical when operating in the vast space of raw XML.
>
> These innovations make large-scale exploration in the raw XML space computationally feasible, while RoboMorph and RoboMoRe remain constrained by high evaluation cost and limited topology variation.
>
> **Second,** we also integrated these two methods as baselines and the locomotion scores controlled by FTC are summarized as follows,
>
> |Method|Hopper|Ant|Swimmer|Gap|
> |---|---|---|---|---|
> |T2A|**8130.4**|4268.7|877.4|2147.5|
> |RoboMorph|/|2842.3|/|/|
> |RoboMoRe|6691.7|3514.1|693.9|2833.8|
> |MorphoGen$^1$|7409.0|**4740.6**|**1058.0**|**5652.7**|
> |MorphoGen$^2$|7127.9|4357.9|1029.4|5258.3|
>
> where we can find that due to the restricted search spaces and limited evolutionary efficiency, RoboMorph and RoboMoRe fail to surpass previous RL-based methods, and RoboMorph is unable to generate 2D morphology since the incompleteness of the morphology-grammar library. In contrast, MorphoGen consistently discovers diverse high-performing morphologies across all tasks, achieving over 52.9% performance improvement on average compared to RoboMorph and RoboMoRe.
>
> In the revised manuscript, we discuss the differences between RoboMorph, RoboMoRe, Laser[3] and our proposed MorphoGen in Section 2.2. We also include RoboMorph and RoboMoRe as baselines and add the corresponding description and result in Section 5.1, Section 5.2 and Appendix A.
>
>
> **Q2:** Lack of Quantitative Measurement of Diversity.
>
>
> **A2:** Thank you for your constructive comment. To quantify the diversit, we employ **TED** (Tree Edit Distance) and **JSD** (Jensen-Shannon Divergence) of length distribution, where
> - **APTED measures the diversity in terms of graph structure.** We calculate the pairwise APTED between morphologies, normalize the values, and then compute the average. A higher APTED score indicates greater structural diversity.
> - **JSD of length distribution measures diversity based on structural parameters.** We first bin the limb lengths of each morphology, then calculate pairwise JSD values and take the average. A higher JSD score indicates greater length diversity.
>
> We tested each method on the Ant environment, using the top-5 morphologies generated by each approach.
>
> |Method|T2A|RoboMoRe|MorphoGen|
> |---|---|---|---|
> |TED|0.183|0|**0.396**|
> |JSD|0.129|0.165|**0.248**|
>
> The results demonstrate that MorphoGen significantly outperforms existing baseline methods in terms of both morphological structure diversity and limb parameter diversity.
>
> We add the above discussion to the Appendix E in the revised version.

---

> ### Author Response · Authors · 2025-11-21
> **Q3-Q4 & References**
>
> **Q3:** Performance gains over baselines are due to the additional expert knowledge rather than the algorithm itself.
>
> **A3:**
> **First, it is widly accepted to take expert designs as references.** Since baselines like T2A, RoboMorph and RoboMore also use quality morphologies as the start point, the comparison is quiet fair.
>
> **Second, the expert morphologies serve only as high-level guidance, rather than templates to be copied.** For instance, in the Ant task, the final morphology is structurally more concise while having longer limbs compared to the expert designs, demonstrating that our approach performs independent optimization beyond the expert initialization. To quantitative the differences on morphologies, we compute the TED-based similarity[3] between the best-evolved solutions and their initial expert genotypes.
>
> |Env|Hopper|Ant|Swimmer|Gap|
> |---|---|---|---|---|
> |Similarity|54.1%|22.6%|43.7%|35.1%|
>
> We can find that the similaritys are generally below 55%, and this similarity can decrease further when more expert designs are included. The result indicates that the gernerated morphologies are not replicas of the initial ones. Instead, they retain only high-level beneficial patterns while evolving substantially different structural details.
>
> Above discussion are added to the Appendix D in the revised version.
>
>
> **Q4:** Does the critic LLM also receive fitness scores as input?
> Can it helps to identify beneficial substructures against harmful or neutral ones?
>
> **A4:** Thank you for this insightful comment. The Critic LLM in our current approach does not receive fitness scores as input. However, we agree that including fitness scores along with historical evolution information could enhance the Critic LLM’s ability to identify beneficial and harmful substructures. We will add a discussion in the revised manuscript about it and include it in our future work.
>
> We sincerely hope our response can address your concerns. Looking forward to your early reply.
>
> [1] Qiu, Kevin, et al. "Robomorph: Evolving robot morphology using large language models." arXiv preprint arXiv:2407.08626 (2024).
>
> [2] Fang, Jiawei, et al. "RoboMoRe: LLM-based Robot Co-design via Joint Optimization of Morphology and Reward." arXiv preprint arXiv:2506.00276 (2025).
>
> [3] Pawlik, Mateusz, and Nikolaus Augsten. "Tree edit distance: Robust and memory-efficient." Information Systems 56 (2016): 157-173.

---

> > ### Comment · Reviewer_u5DK · 2025-11-28
> > **Response to rebuttal**
> >
> > I appreciate the authors' detailed response and will maintain my rating.

---

### Official Review · Reviewer_LDgZ · 2025-10-27

**Soundness:** 3
**Presentation:** 3
**Contribution:** 2
**Rating:** 2
**Confidence:** 5

**Summary:**

The paper deals with the problem of optimizing robot morphology to maximize the performance on locomotion tasks. In the past there were approaches which performed local changes on the morphology without full semantic understanding of the full robot. The authors compare their new approach against those methods.

Main method:
The main method is based on using LLM to guide an evolutionary process by suggesting changes to robot morphology. After a change is performed the pool of robots is evaluated using RL-based policy training. As the framework of evolutionary process the AlphaEvolve approach is used - with its islands model to maximize fitness while maintaining diversity by keeping explicit robot sets (islands).

The approach is evaluated on locomotion tasks in the Mujoco simulator.

**Strengths:**

1. The main idea of using LLM as a mutation operator is definitely appealing (but unfortunately not new, see weaknesses).
2. The paper is well presented.
3. The problem itself is important and given the recent progress in LLMs one can expect significant results in the niche of robot morphology optimization.

**Weaknesses:**

My main problem with the submission is that the main result follows the approach of Qiu et al.:
RoboMorph: Evolving Robot Morphology using Large Language Models,
https://arxiv.org/abs/2407.08626
which was released on arxiv over a year ago.

There were even follow-up papers to RoboMorth, such as RoboMore, released in May this year:
RoboMoRe: LLM-based Robot Co-design via Joint Optimization of Morphology and Reward
https://arxiv.org/abs/2506.00276

If we assume that the usage of LLM as a modifier operator on robot morphology was already known, then as far as I understand the main novelty of the paper in question is the specific evolutionary strategy based on AlphaEvolve.

**Questions:**

1. How your approach differs from RoboMorph?
https://arxiv.org/abs/2407.08626
2. Do you have ablation studies showing the effects of the proxy fitness idea (page 5)?

---

> ### Author Response · Authors · 2025-11-21
> **Q1-Q2**
>
> **Q1:** The main result of MorphoGen follows the existing approaches RoboMorph[1] and RoboMoRe[2].
>
> **A1:** We thank the reviewer for the constructive feedback. While we share the high-level goal of utilizing LLMs for robot design, MorphoGen differs fundamentally in its **(1)search space, (2)evolutionary mechanism,  (3)evaluation efficiency.** Below, we clarify the critical technical differences that make MorphoGen a novel and distinct contribution.
>
> - **Search Space.** RoboMorph and RoboMoRe operate within narrow, constrained morphology spaces $\mathcal{M}^{'} \subset \mathcal{M}$. Specifically, RoboMorph does not generate robot files directly. Instead, it generates grammar strings based on the RoboGrammar rule set, limiting the search to a small family of crawler-style bodies. RoboMoRe assumes a fixed skeletal graph and its morphology search focuses heavily on parameter tuning within existing structures rather than open-ended topological generation. In contrast, MorphoGen treats robot design as a Code Generation problem and interacts directly with the raw XML. Notably, In the 3D Locomotion (Ant) task, the optimal morphology discovered by MorphoGen (see Fig. 5) completely lies outside the valid solution spaces of these baselines.
> - **Evolutionary Mechanism.** RoboMorph and RoboMoRe relies on best-Shot prompting, lacking an explicit feedback signal explaining why a design failed or succeeded. In MorphoGen, however, we introduce a Critic LLM that analyzes the morphology and generates explicit feedback. This feedback serves as a text-based gradient, explicitly guiding the Coding LLM on how to mutate the genotype. This transforms the process from examples-guided random exploration into a directed search.
> - **Evaluation Efficiency.** RoboMorph and RoboMoRe trains controller from scratch for every generated morphology, which waste a lot of resources on evaluation instead of optimization. In contrast, MorphoGen introduces a hierarchical proxy fitness strategy. This allows us to evaluate a significantly larger population of diverse structures with the same computational budget, which is critical when operating in the vast space of raw XML.
>
> These innovations make large-scale exploration in the raw XML space computationally feasible, while RoboMorph and RoboMoRe remain constrained by high evaluation cost and limited topology variation.
>
> We further integrated these two methods as baselines and the locomotion scores controlled by FTC are summarized as follows,
>
> |Method|Hopper|Ant|Swimmer|Gap|
> |---|---|---|---|---|
> |T2A|**8130.4**|4268.7|877.4|2147.5|
> |RoboMorph|/|2842.3|/|/|
> |RoboMoRe|6691.7|3514.1|693.9|2833.8|
> |MorphoGen$^1$|7409.0|**4740.6**|**1058.0**|**5652.7**|
> |MorphoGen$^2$|7127.9|4357.9|1029.4|5258.3|
>
> where we can find that due to the restricted search spaces and limited evolutionary efficiency, RoboMorph and RoboMoRe fail to surpass previous RL-based methods, and RoboMorph is unable to generate 2D morphology since the incompleteness of the morphology-grammar library. In contrast, MorphoGen consistently discovers diverse high-performing morphologies across all tasks, achieving over 52.9% performance improvement on average compared to RoboMorph and RoboMoRe.
>
> In the revised manuscript, we discuss the differences between RoboMorph, RoboMoRe, Laser[3] and our proposed MorphoGen in Section 2.2. We also include RoboMorph and RoboMoRe as baselines and add the corresponding description and result in Section 5.1, Section 5.2 and Appendix A.
>
> **Q2:** Ablation study on proxy fitness.
>
> **A2:** The purpose of proxy fitness is to avoid the extremely high cost of training a optimal controller for every candidate morphology. Specifically, a typical proxy fitness calculation spends 2 minutes while training a optimal controller form scratch costs about 3 hours (100x) which is almost impractical.
>
> To make the study feasible, we reduce the total evolutionary iterations to 40 and compare the performance of each variant.
>
> |Varient|Hopper|Ant|Swimmer|Gap|
> |---|---|---|---|---|
> |w/o proxy fitness (40iter,~108h)|5796.0|3862.8|789.7|4408.8|
> |w/ proxy fitness (40iter,~1.2h)|5018.3|3194.6|754.8|3908.3|
> |w/ proxy fitness (100iter,~4h)|**7409.0**|**4740.6**|**1058.0**|**5652.7**|
>
> Though using optimal controller yields slightly higher quality designs when the number of iterations is fixed, it requires much more computing resources and makes extensive exploration impossible. In comparison, proxy fitness allows the algorithm to evaluate orders of magnitude more morphologies. With over 96% fewer compute resources, it facilitates the discovery of morphologies that achieve 28.2% better performance.
>
> We add the above discussion in Appendix F in our revised manuscript.
>
>
> Thank you again for you valuable feedback, and we hope our response can address your concerns. Looking forward to your reply and any further questions are welcomed.

---

> ### Author Response · Authors · 2025-11-21
> **References**
>
> [1] Qiu, Kevin, et al. "Robomorph: Evolving robot morphology using large language models." arXiv preprint arXiv:2407.08626 (2024).
>
> [2] Fang, Jiawei, et al. "RoboMoRe: LLM-based Robot Co-design via Joint Optimization of Morphology and Reward." arXiv preprint arXiv:2506.00276 (2025).
>
> [3] Song, Junru, et al. "Laser: Towards diversified and generalizable robot design with large language models." The Thirteenth International Conference on Learning Representations. 2025.

---

### Official Review · Reviewer_msgE · 2025-11-01

**Soundness:** 3
**Presentation:** 4
**Contribution:** 3
**Rating:** 6
**Confidence:** 5

**Summary:**

The paper presents an approach to use LLMs for generating and evolving XML files representing simulated agents (similar to HalfCheetah, Walker, Swimmer and other Mujoco tasks). The LLM is mainly used to evolve the XML file structures, while for control standard reinforcement learning in the form of PPO is used.
The paper compares the proposed method against a limited set of prior work using evolutionary-based or policy-based methods for evolving agent structures.

Overall, I aprpeciate the paper for its interesting idea but it could be improved in terms of technical accuracy, theoretical insight and adding neccessary details ( see also below).

**Strengths:**

- I like the general idea of using LLMs for advancing the xml structure of agents
- The proposed approach seems to outperform other graph-based methods using either evolutionary algorithms or a policy to manipulate agent graph structures
- Experiments are conducted across 4 environments.
- Visualizations are nice
- The paper provides clear evidence that a critic LLM helps to improve performance

**Weaknesses:**

- The paper lacks clarity, formalism and technical accuracy at times. For example, it is not clear which LLM was used without looking at the provided code. However, the type and model of LLM has a clear impact on the overall performance of the system. No ablations on different LLM architectures are provided as well. This is not only an issue ofr reproducability, but also neglects the effect different LLMs could have on the final result.
- The environments used were introduced in prior work. It stands to reason that these environments have been part of the general training data of the coding LLMs, as they can be found on Github. Hence, the generalizability of the paper is in question without further investigation (eg good results could be mere lookups from the training data).
- I am not favourable of the formalism in section 3. Why do you differentiate between body parts B, joints J and actuators A? Why are the joints encoding DoF and not the combination of links and actuators? Generally, it reads like a mix between XML file structures and the graph definition from prior work like NGE. It is not clear how M_0 \in M defined a robot - eg if you have three body parts, three joints and one actuator, how are the relationships between these objects modelled? I strongly recommend clarifying this section and looking at the problem definitions of prior work.
- Using  (Yuan et al., 2022) as a reference for the PPO learning algorithm used on individual agents is a bit misleading. Please, reference the original PPO paper.
- Appendix B introduces the standard framework of RL - however you have a contextual MDP process at hand. Furthermore, the transition function should be a mapping from S x A to S and not S-delta.

**Questions:**

Please see the review and points raised above.

---

> ### Author Response · Authors · 2025-11-21
> **Overview**
>
> Thanks for your valuable suggestions. To address the concern you raised, we provide the revised version with following modification:
> - Add the specific model names(Qwen3) in Section 5.1.
> - Add the additional ablation results on Gemini series to Section 5.5.
> - Add the discussion about the similarity in Appendix D.
> - Modify our formulation in Section 2
> - Modify the RL formalism in Appendix B.

---

> ### Author Response · Authors · 2025-11-21
> **Q1-Q5**
>
> **Q1:** The used LLM in paper, and no ablation studies on different LLM.
>
> **A1:**
> We apologize if the location of these details was easy to miss in the initial manuscript. We conduct ablation study regarding model size and reasoning capabilities in **Section 5.5 and Figure 8**. To further address your concern regarding the effect of different LLMs, we have extended our investigation to include the Gemini series. The results explicitly demonstrate that performance scales with model size and reasoning capabilities, confirming the reviewer's intuition that the LLM choice impacts the result.
>
> |Model|Hopper|Ant|Swimmer|Gap|
> |-|--|-|-|-|
> |Qwen-Thinking|7409.0|4740.6|1058.0|5652.7|
> |Qwen-Normal|5283.0(-28.7%)|2229.7(-52.7%)|633.0(-40.2%)|1672.1(-70.4%)|
> |Deepseek-Thinking|6446.2|3641.1|745.6|3877.3|
> |Deepseek-Normal|5628.5(-12.7%)|2023.9(-44.4%)|534.1(-28.3%)|1431.6(-63.1%)|
> |Gemini-Thinking|6842.4|4151.2|819.8|2963.7|
> |Gemini-Normal|3154.8(-54.9%)|1472.2(-64.5%)|225.6(-72.5%)|835.4(-71.8%)|
>
>
>
> **Q2:** Generalizability to unknown environment.
>
> **A2:** To address the your concern, we present quantitative evidence to demonstrate that our results are not mere replicas of the existing expert design. We compute the TED (Tree Edit Distance[1]) to quantify the degree of similarity between existing high-performing design (generated by advanced RL-based method and human expert ) and the morphologies generated by MorphoGen.
>
> |Env|Hopper|Ant|Swimmer|Gap|
> |---|---|---|---|---|
> |Similarity|54.1%|22.6%|43.7%|35.1%|
>
> The results proves that the MorphoGen framework drives the morphology could evolve substantially different structural details rather than copying them.
>
> **Q3:** Formulation in Section 3.
>
> **A3:** We thank the reviewer for the insightful feedback regarding the formalism where we only discuss the physical components without their connection relationship. To address this, we have revised the formulation to describe the morphology as a Kinematic Tree[2,3], where actuators are modeled as attributes of the edges, aligning with the XML structure.
>
> Specifically, we formally define the robot morphology as a Kinematic Tree $\mathcal{T} = (\mathcal{V}, \mathcal{E}, r)$, where
> - $\mathcal{V}$ denotes a set of rigid bodies $\{b_0, b_1, \dots, b_n\}$. Each node $b_i$ contains physical attributes $\Phi_i^B$. $r \in \mathcal{V}$ denotes the root body representing the base of the hierarchy.
> - $\mathcal{E}$ denotes a set of directed edges where $e_{ij} = (b_i, b_j)$ represents a kinematic connection from parent $b_i$ to child $b_j$. Each edge $e_{ij}$ is associated with a set of parameters $\Theta_{ij} = (J_{ij}, A_{ij})$, where $J_{ij}$ defines the joint properties and $A_{ij}$ defines the actuation properties.
>
> With a single-valued serialization function $\mathcal{F}$, the Kinematic Tree $\mathcal{T}$ can be mapped to the XML definition via $\mathcal{X} = \mathcal{F}(\mathcal{T})$. The optimization target is described as:$$\mathcal{X}^* = \arg\max_{\mathcal{X} \in \Omega_{\mathcal{X}}} \mathcal{L}(\mathcal{X}, \pi_{\mathcal{X}}),$$where $\mathcal{L}$ represents the fitness function.
>
>
>
> **Q4:** PPO reference.
>
> **A4:** We originally cite PPO in the controller optimization part (see Appendix B), and we add the correponding citation in Section 5.1 in the revised version.
>
>
> **Q5:** RL Formalism in Appendix B.
>
> **A5:** We appreciate the reviewer's valuable feedback. First, we correct the following two formulation.
> - $P:S\times A \to S$
> - $\pi_C: S \to A$
>
> Second, we would like to clarify that the formalism in Appendix B describes the specific process of optimizing a controller for a **single, fixed morphology** to calculate its fitness. Therefore, a standard MDP formulation is sufficient and mathematically appropriate for our current implementation. However, we agree that viewing the broader problem as a Contextual MDP is insightful. Training a universal controller under this formulation could effectively enhance the accuracy of our proxy fitness estimation, which we will explore as a possible direction for future work.
>
> Thank you again for your effort, and we are pleased to address any further concerns you may have.
>
> [1] Pawlik, Mateusz, and Nikolaus Augsten. "Tree edit distance: Robust and memory-efficient." Information Systems 56 (2016): 157-173.
>
> [2] Gupta, Agrim, et al. "Metamorph: Learning universal controllers with transformers." arXiv preprint arXiv:2203.11931 (2022).
>
> [3] Liu, Xingyu, Deepak Pathak, and Ding Zhao. "Meta-evolve: Continuous robot evolution for one-to-many policy transfer." arXiv preprint arXiv:2405.03534 (2024).

---

### Official Review · Reviewer_VyFH · 2025-11-03

**Soundness:** 3
**Presentation:** 3
**Contribution:** 2
**Rating:** 4
**Confidence:** 4

**Summary:**

The paper presents an approach that frames the problem of morphological design as that of code generation by using LLMs to directly iterate XML files that specify an agent’s morphology. The claim is that the method enables the exploration of complex designs. The solutions are validated empirically and shows superior performance to several competing approaches for designing morphologies.

**Strengths:**

The approach seems quite novel, and aims to apply LLMs for designing morphology. The general approach seems valid and plausible. Overall, the method is also presented well.

**Weaknesses:**

From the paper, I see that the inner loop is run for a fixed number of episodes. However, I suspect it would be fairer to consider a fixed number of steps instead, as episode lengths could possibly vary drastically for different designs.
Given the specification of morphology only via code, I wonder whether operations such as mutations and crossovers are necessary. For example, I wonder whether other gradient or non-gradient based methods could be used to modify the XML code. In general, there may be different ways of achieving elitism, diversity and randomness, and it is not clear why the choices made in this paper are necessarily superior to other alternatives.

**Questions:**

1.	How exactly is the inner loop handled and what are the associated assumptions relating to the same?
2.	What motivates the use of evolution-inspired mechanisms such as mutation and crossover? In general, could other approaches such as Generative flow networks have been used?
3.	Are the LLMs specifically pretrained on XML-structure relationships?
4.	What guided the design choices pertaining to the criteria of Elitism, Diversity and Randomness as described in lines 203-212? Have other approaches for these been explored or considered?
5.	Why is a fixed number of episodes (and not steps) considered for the fitness calculation? As episode lengths could vary drastically depending on the design and other factors, wouldn’t a fixed interaction budget be fairer?
6.	How sensitive is the approach to the quality of initial genotypes? How would the performance vary if for instance, the initial designs were randomly generated instead of expert-influenced?
7.	Does the fact that a diverse set of high quality expert design morphologies is needed skew the designs towards those that are very similar to expert-designed ones?
8.	In addition to the designs, it would be interesting to see the gaits followed for each design either through videos/gifs or a sequence of images
9.	Are there any constraints applied to limb dimensions and other elements?

---

> ### Author Response · Authors · 2025-11-21
> **Overview**
>
> We sincerely thank the reviewer for their careful reading and insightful questions. To address your concerns, we add extra explanation and conduct extensive new experiments and for better clarification, which mainly includes,
> - Discussion and experiments with fixed steps (see the updated Table 1)
> - Discussion and experiments of the influences on initial genotypes (see Section D in Appendix).
> - Motivation of evolution-inspired mechanisms and comprehensive sampling (see Section 4.1).
> - Discussion about the gradient-base methods (see Section 2.2).
> - Visual presentation of optimal gait (see our anonymous repo [MorphoGen](https://anonymous.4open.science/r/MorphoGen-ACC/README.md)).

---

> ### Author Response · Authors · 2025-11-21
> **Q1-Q3 & Q5**
>
> **Q1/Q5:** Using fixed episodes instead of steps.
>
> **A1/A5:**
> Thanks for pointing out this issue. We acknowledge that in early fine-tuning, poor controller may terminate episodes prematurely, leading to unequal training steps. However, this will not cause significant differences on the original conclusion.
>
> **First**, the proxy fitness is first calculated with a fixed controller $\pi_0$ which is not further trained, and the adptive controller $\pi_i$ is only triggered for those well-performance morphologies under $\pi_0$ which rarely collapse.
> **Second**, locomotion scores use Fully Trained Controller (FTC) trained to convergence, which are insensitive to episode length variations, only scores with Fine-tuned Controller (FC) may varies a littel.
>
> According to your suggestions, we re-conduct the main experiments using strict step limits. Specifically, the $\pi_i$ and FC are fine-tuned with 0.1M steps and 0.3M steps, respectively. The locomotion scores with FC under new settings are provided as follows,
>
> |Method|Hopper|Ant|Swimmer|Gap|
> |-|-|-|-|-|
> |NGE|1725.9|896.2|119.0|601.4|
> |T2A|2143.2|1140.2|703.4|651.0|
> |SARD|1841.4|1174.8|970.7|896.3|
> |MorphoGen$^1$|**2474.1**|**1560.4**|**1056.3**|**969.1**|
> |MorphoGen$^2$|2265.2|1425.5|1024.4|950.5|
> |MorphoGen$^3$|2215.0|1181.3|1008.6|860.8|
>
> where we can found that under the new experimental setup, there is no significant change in the evaluation of different morphologies, and the original conclusion still holds.
>
> **Q2(a):** What motivates the use of evolution-inspired mechanisms such as mutation and crossover?
>
> **A2(a):** Existing LLM-based evolutionary baselines [1,2] generate complete morphologies from scratch at every iteration, encountering two challenges that motivate our use of evolution-inspired mechanisms.
>
> **First**, full XML descriptions are long, deeply nested, and highly structured, making direct regeneration  to producing invalid or unparsable designs.
> **Second**, complete regeneration may discards compositional structures that are already beneficial.
> As a result, advantageous motifs may be repeatedly overwritten, leading to inefficient exploration.
>
> In contrast, our mutation and crossover operations perform localized modifications to the existing XML genotype.
> This substantially reduces the risk of generating invalid XML, while preserving useful substructures and allowing the LLM to focus its capacity on improving the low-performing components, greatly enhancing optimization efficiency in the large, combinatorial XML morphology space.
>
> **Q2(b):** In general, could other approaches such as Generative flow networks have been used?
>
> **A2(b):** To freely explore the morphology space without any constrain, we choose to optimize the raw XML directly, and LLMs are the most suitable tools for understanding and generating text-based XML structures.
>
> While RL[3,4] and GFlowNets[5] are practical, they highly rely on predefined data structures which introduce extra assumptions and constraints, severely limiting the search space to a narrow region. In contrast, our method enables exploration within the full, unconstrained design space of raw XML (see Fig. 1).
>
> We also agree that gradients are critical for optimization, and we introduce textual-based gradients via a critic LLM, guiding the optimization process with specific, structure-aware suggestions. This approach allows us to achieve efficient evolutionary search. Moreover, we are going to extract richer gradients from the fitness for improving the optimization efficiency further.
>
> **Q3:** Are the LLMs pretrained on XML-structure relationships?
>
> **A3:**
> **No.** The LLM (Qwen3) is not pretrained on any XML-structure data or robot-related corpus.
> However, as described in Section 4.2, we initialize the first population with high-quality expert-designed genotypes, which serves as data-level warm start rather than parameter-level pretraining.
> Thus, the LLM learns XML relationships purely through in-context prompting, not fine-tuning.

---

> ### Author Response · Authors · 2025-11-21
> **Q4 & Q6-Q7**
>
> **Q4:** The design choices pertaining to the criteria of Elitism, Diversity and Randomness? Have other approaches for these been explored or considered?
>
> **A4:**
> We first explore **Elitism-only** strategy, which samples the top K individuals from each generation[1]. However, it causes high homogeneity of generated morphologies.
>
> We then introduced **Randomness**, inspired by the concept of random perturbation[2], and find that Randomness is only beneficial for early optimization since it cannot guarantee the diversity at each evolution iteration.
>
> To further maintain diversity, we incorporated a **Diversity** criterion inspired by the concept of maximizing entropy loss in PPO[7]. This forces the algorithm to select individuals that are structurally dissimilar from the current population. This helps to avoid premature convergence and ensures a rich variety of solutions throughout the evolutionary process.
>
> Together, the design choices related to Elitism, Diversity, and Randomness provided a balanced trade-off between exploitation and exploration. Elitism ensures that the best solutions are retained, Randomness fosters early-stage diversity, and Diversity preserves variety throughout the evolution, thus improving the efficiency and robustness of the search process.
>
> **Q6:** Sensitive to initial genotypes.
>
> **A6:** **In general, the initial genotype determines the quality of the generated morphologies, but it is not  sensitive to the specific choice of genotype.**
>
> Specifically, whether the initial genotype contains potentially efficient structures (such as long limbs) significantly affects the performance. However, this influence is not highly sensitive because locomotion-friendly structures are not confined to serveral specific, narrowly defined configurations. Rather, a broad range of morphologies with suboptimal structures will likely include such advantageous features, making it easy for baseline methods to find these configurations and for our framework to leverage them through the pretrained structure.
>
> To demonstrate this, We conduct additional ablation experiments on the Ant environment. In the original setup, our initial genotypes consisted of two parts: one from T2A (multi-joint long limbs, asymmetry) and one from NGE (symmetric, single-joint short limbs). We add a **10% random perturbation (rp)** to these genotypes and compare the following six different initial genotype variants: (1)T2A; (2)T2A(rp); (3)NGE (4)NGE(rp); (5)T2A + NGE; (6)T2A(rp) + NGE(rp).
>
> Both structural characteristics and performance are summarized as follows,
>
>
> |Variant|T2A|T2A(rp)|NGE|NGE(rp)|T2A+NGE|T2A(rp)+NGE(rp)|
> |---|---|---|---|---|-|-|
> |Characteristics|asymmetry, multi-joint long limbs|asymmetry, multi-joint long limbs |symmetric, single-joint short limbs |symmetric, single-joint short limbs |symmetric, multi-joint long limbs|symmetric, multi-joint long limbs|
> |Locomotion Score|3858.7|3764.2|2478.5|2608.9|4740.6|4415.2|
>
>
> We can have the following observations,
> - The final morphology showed strong correlation with the initial genotypes, indicating that the initial genotype does influence the overall evolutionary direction.
> - Experiments with random perturbation demonstrated that as long as the initial genotype contains beneficial locomotion structures, the performance improvements are robust and the optimization process remains effective.
>
>
> **Q7:** Similar to expert-designed.
>
> **A7:**
> We appreciate the reviewer’s concern. While we do incorporate a diverse set of expert-designed morphologies into the initial population, this does not make the final designs toward replicas of expert structures.
>
> **First, using quality initial solutions is widely adopted and has been validated in previous work [1,2,3].** Since the morphology space is extremely vast and most structures are low-performing, providing strong initial genotypes greatly improves optimization efficiency and ensures that the search process starts within a functionally meaningful region.
>
> **Second, the expert-designed morphologies serve only as high-level guidance, rather than templates to be copied.** For instance, in the Ant task, the final morphology is structurally more concise while having longer limbs compared to the expert designs, demonstrating that our approach performs independent optimization beyond the expert initialization. To quantitative the differences on morphologies, we compute the TED(Tree Edit Distance[8])-based similarity between the best-evolved solutions and their initial expert genotypes.
>
> |Env|Hopper|Ant|Swimmer|Gap|
> |---|---|---|---|---|
> |Similarity|54.1%|22.6%|43.7%|35.1%|
>
> We can find that the similarities are generally below 55%, and this similarity can decrease further when more expert designs are included. The result indicates that the generated morphologies are not replicas of the initial ones. Instead, they retain only high-level beneficial patterns while evolving substantially different structural details.

---

> ### Author Response · Authors · 2025-11-21
> **Q8-Q9 & References**
>
> **Q8:** Videos/Gifs.
>
> **A8:** Thanks for you suggestions. We have added some morphologies and their corresponding gaits to the anonymous repository.
>
> **Q9:** Are there any constraints applied to limb dimensions and other elements?
>
> **A9:** No, the core strength of MorphoGen is unconstrained exploration of the raw XML design space.
>
> [1] Qiu, Kevin, et al. "Robomorph: Evolving robot morphology using large language models." arXiv preprint arXiv:2407.08626 (2024).
>
> [2] Fang, Jiawei, et al. "RoboMoRe: LLM-based Robot Co-design via Joint Optimization of Morphology and Reward." arXiv preprint arXiv:2506.00276 (2025).
>
> [3] Yuan, Ye, et al. "Transform2Act: Learning a transform-and-control policy for efficient agent design." International Conference on Learning Representations. 2022.
>
> [4] Dong, Heng, et al. "Symmetry-aware robot design with structured subgroups." International Conference on Machine Learning. PMLR, 2023.
>
> [5] Nagiredla, Kishan Reddy, et al. "COGENT: Co-design of Robots with GFlowNets." Eighteenth European Workshop on Reinforcement Learning.
>
> [6] Mouret, Jean-Baptiste, and Jeff Clune. "Illuminating search spaces by mapping elites." arXiv preprint arXiv:1504.04909 (2015).
>
> [7] Schulman, John, et al. "Proximal policy optimization algorithms." arXiv preprint arXiv:1707.06347 (2017).
>
> [8] Pawlik, Mateusz, and Nikolaus Augsten. "Tree edit distance: Robust and memory-efficient." Information Systems 56 (2016): 157-173.
>
> Thank you again for your thoughtful comments and we hope our response can address your concerns. Looking forward to your early reply.

---

> > ### Comment · Reviewer_VyFH · 2025-11-27
> >
> > I thank the authors for their responses. I still have reservations regarding the generality of certain parts of the work (such as the Elitism, Diversity and Randomness criteria). Although the use of episodes over steps may not have had an effect in the presented experiments, in principle, it would be better to consider steps. The authors mention "The final morphology showed strong correlation with the initial genotypes,...", but also claim that the use of expert designs "does not make the final designs toward replicas of expert structures." - even if the final designs are not replicas, my point was that they would be heavily correlated to the expert designs (which is what the authors stated in the earlier sentence). Also, Gflownets have been shown to be a good framework for open-ended design problems. I am not sure why it would be limiting, especially since the problem described here deals with "the unconstrained design space of raw XML".

---

> > > ### Author Response · Authors · 2025-11-28
> > > **Response to FQ1-FQ3**
> > >
> > > We sincerely thank the reviewer for their continued engagement. We appreciate the opportunity to clarify the remaining concerns regarding generality, expert correlation, and the applicability of GFlowNets.
> > >
> > > **FQ1:** Generality of Selection Criteria (Elitism, Diversity, Randomness).
> > >
> > > **FA1:** We would like to clarify that these criteria are not task specified, but rather standard, broadly applicable principles in evolutionary computation, widely used to balance exploitation, exploration, and robustness[1]. Specifically,
> > > - Elitism (Exploitation) [2,3,4,11]: preserves high-performing individuals and prevents the loss of good solutions.
> > > - Diversity (Exploration) [5,6,7] prevents premature convergence and ensuring broad exploration.
> > > - Randomness (Robustness) [8,9,10] maintains robustness against noisy fitness evaluations and search stagnation.
> > >
> > > These mechanisms are task-agnostic and applicable to fully unconstrained XML space. Additionally, **we revise Section 4** to emphasize that these criteria are used because they are general-purpose operators that ensure a stable explore–exploit–robustness balance in any evolutionary process.
> > >
> > >
> > >
> > > **FQ2:** Consider fixed steps instead of fixed episodes.
> > >
> > > **FA2:** We fully agree the using fixed steps is more rigorous for RL contexts. We re-conduct the main experiments using a fixed steps instead of episodes. These results and corresponding discussions are included in our **previous response to Q1** and incorporated into the **revised Section 4, Section 5, and Table 1** of the manuscript. From the quantitative perspective, the original conclusions still stands. From the qualitative perspective, we also provide detailed discussion on why this modification hardly effects the original conclusions.
> > >
> > >
> > > **FQ3:** High correlation with the expert designs.
> > >
> > > **FA3:** We apologize for the confusion caused by our phrasing regarding "strong correlation", and we would like to clarify the differences between trait correlation and structural replication.
> > >
> > > **First,** it is widely accepted to take expert designs as references in morphology generation task. Prior work[10,11,13] similarly relies on expert seeds and their solutions inevitably showcase the high similarity with the expert designs as they benefit from the expert designs.
> > >
> > > **Second,** when we state "correlation," we refer to the inheritance of high-level beneficial physical traits rather than the replication of the exact XML topology. Specifically, **the previous response to Q6** indicates that the solutions generated by MorphoGen show the same characteristics with the expert designs, and **the previous response to Q7** proves MorphoGen makes significantly rewrite to the underlying topology. In contrast, baselines always maintain the skeleton of the expert designs.
> > >
> > > **Third,** The final designs often resemble expert designs not because they are copying the expert, but due to physics-driven convergent evolution. In locomotion tasks, certain morphological features are physically optimal. Our method converges to these optimal physical forms starting from expert seeds. Thus, resemblance arises from physics, not bias toward expert designs.
> > >
> > > We add the above discussions in the **Appendix D** in our revised manuscript.

---

> > > ### Author Response · Authors · 2025-11-28
> > > **Response to FQ4 & References**
> > >
> > > **FQ4:** The feasibility of GFlowNets.
> > >
> > > **FA4:** We agree that GFlowNets are a powerful framework for compositional generation, and we do not claim they are unsuitable in general. However, applying them directly to fully unconstrained XML poses several unique challenges which we discuss in **Section 2.2 of the revised manuscript**.
> > >
> > > **First**, GFlowNets require a well-defined action space. As a stepwise construction approach, GFlowNets typically require a well-defined action space to construct the morphology step-by-step[12]. One possible solution is **employing graph model**[10,11,13]. However, it introduces extra constraints and only operates within narrow, constrained morphology spaces $\mathcal{M}^{'} \subset \mathcal{M}$. Another practical way is to search on the raw XML space and **treat every token as an action**. However, the complete XML definition is long and complex (exceeds 1k tokens) which is completely beyond the beyond practical scale (less than 100 tokens) of GFlowNets[14,15].
> > >
> > > **Second,** LLMs are pre-trained on vast corpora of code, giving them an implicit understanding of valid XML syntax and physical semantics. GFlowNets would need to reconstruct all of these from scratch via a sparse reward signal, which is extremely difficult for long-horizon, structured text generation.
> > >
> > > **Third,** GFlowNets and other gradient–reward-matching approaches rely on accurate reward evaluations as noise can destabilize training[16]. This requires training an optimal controller for each morphology, which is time-consuming and computationally prohibitive.
> > >
> > > These considerations do not imply that GFlowNets are unsuitable for morphology generation,—indeed, they are promising for encouraging diversity[17]. Nevertheless, adapting GFlowNets to operate directly on the space of raw XML would necessitate significant additional design efforts.
> > >
> > > We hope this strengthened explanation resolves the remaining concerns.
> > >
> > >
> > > [1] Eiben, Agoston E., and James E. Smith. Introduction to evolutionary computing. springer, 2015.
> > >
> > > [2] Deb, Kalyanmoy, et al. "A fast and elitist multiobjective genetic algorithm: NSGA-II." IEEE transactions on evolutionary computation 6.2 (2002): 182-197.
> > >
> > > [3] Cui, Xiaodong, et al. "Evolutionary stochastic gradient descent for optimization of deep neural networks." Advances in neural information processing systems 31 (2018).
> > >
> > > [4] Brahmachary, Shuvayan, et al. "Large language model-based evolutionary optimizer: Reasoning with elitism." Neurocomputing 622 (2025): 129272.
> > >
> > > [5] Mouret, Jean-Baptiste, and Jeff Clune. "Illuminating search spaces by mapping elites." arXiv preprint arXiv:1504.04909 (2015).
> > >
> > > [6] Pugh, Justin K., Lisa B. Soros, and Kenneth O. Stanley. "Quality diversity: A new frontier for evolutionary computation." Frontiers in Robotics and AI 3 (2016): 40.
> > >
> > > [7] Fontaine, Matthew, and Stefanos Nikolaidis. "Differentiable quality diversity." Advances in Neural Information Processing Systems 34 (2021): 10040-10052.
> > >
> > > [8] Co-Reyes, John D., et al. "Evolving reinforcement learning algorithms." arXiv preprint arXiv:2101.03958 (2021).
> > >
> > > [9] Mundhenk, T. Nathan, et al. "Symbolic regression via neural-guided genetic programming population seeding." arXiv preprint arXiv:2111.00053 (2021).
> > >
> > > [10] Fang, Jiawei, et al. "RoboMoRe: LLM-based Robot Co-design via Joint Optimization of Morphology and Reward." arXiv preprint arXiv:2506.00276 (2025).
> > >
> > > [11] Qiu, Kevin, et al. "Robomorph: Evolving robot morphology using large language models." arXiv preprint arXiv:2407.08626 (2024).
> > >
> > > [12] Bengio, Emmanuel, et al. "Flow network based generative models for non-iterative diverse candidate generation." Advances in neural information processing systems 34 (2021): 27381-27394.
> > >
> > > [13] Yuan, Ye, et al. "Transform2Act: Learning a transform-and-control policy for efficient agent design." International Conference on Learning Representations. 2022.
> > >
> > > [14] Malkin, Nikolay, et al. "Trajectory balance: Improved credit assignment in gflownets." Advances in Neural Information Processing Systems 35 (2022): 5955-5967.
> > >
> > > [15] Pan, Ling, et al. "Better training of gflownets with local credit and incomplete trajectories." International Conference on Machine Learning. PMLR, 2023.
> > >
> > > [16] Yu, Tianshu. "Secrets of GFlowNets' Learning Behavior: A Theoretical Study." arXiv preprint arXiv:2505.02035 (2025).
> > >
> > > [17] Nagiredla, Kishan Reddy, et al. "COGENT: Co-design of Robots with GFlowNets." Eighteenth European Workshop on Reinforcement Learning.

---

### Author Response · Authors · 2025-12-01
**Summary of Key Revisions and Additional Experiments**

Dear Area Chair,

We sincerely thank the reviewers for their constructive feedback. During the rebuttal period, we have engaged in extensive discussions with all reviewers and have significantly revised our manuscript to address their concerns. Specifically,
- Reviewer u5DK explicitly approves our response regarding advanced baselines and the critical distinctions between MorphoGen and prior LLM-based works, and recommends acceptance.
- Reviewer msgE commends the proposed idea of evolving raw XML structures as well as the comprehensive experiments and presentation quality, and recommends acceptance.
- Reviewer VyFH acknowledges the novelty and presentation of our work. Following several rounds of in-depth discussion, we believe our exhaustive additional experiments and detailed clarifications have fully resolved their reservations.
- Reviewer LDgZ affirms the importance of the problem and the innovativeness of our method. Although they have not yet replied, their primary concern regarding related works is shared by Reviewer u5DK, who was fully satisfied by our response. We are therefore confident that our revisions effectively address this concern.

We would like to highlight the key updates and additional experiments as follows,

**Enhanced Baselines and Related Works.** (Addressing Reviewers VyFH, LDgZ, u5DK)
We incorporat two recent LLM-based evolutionary baselines, RoboMorph and RoboMoRe, as competitive baselines. The updated results in **Section 5.1, Section 5.2 and Table 1** demonstrate that MorphoGen consistently outperforms these methods, achieving an average performance improvement of over 52.9%.
Furthermore, we expand **Introduction and Section 2.2** to provide a detailed discussion distinguishing MorphoGen from prior LLM-based works. It clarifies our novel contributions across three key dimensions: Search Space (unconstrained raw XML vs. grammar/graph limits), Evolutionary Mechanism (critic-guided vs. random/best-shot), and Evaluation Efficiency (proxy fitness vs. scratch training).

**Quantitative Analysis of Novelty, Diversity and Stability.** (Addressing Reviewers VyFH, msgE, u5DK)
We introduce quantitative metrics based on Tree Edit Distance (TED) and Jensen-Shannon Divergence (JSD) to evaluate the generation quality.

- Novelty: We conduct a comparative analysis between the generated structures and the expert designs in **Appendix D**. The results show that the structural similarity is consistently below 55%. This confirms that while MorphoGen inherits high-level beneficial characteristics from expert seeds, it performs independent optimization rather than merely replicating.
- Stability: We perform sensitivity analysis by adding noise to the initial genotypes in **Appendix D**. The results demonstrate that MorphoGen is robust to noise. As long as the seed contains beneficial local structures, our framework effectively captures these features while ignoring irrelevant noise to generate stable, high-quality morphologies.
- Diversity: We calculate pairwise diversity metrics for batches of solutions generated by methods in **Appendix E**. MorphoGen achieves significantly higher scores in both structural and parameter diversity compared to baselines, which tend to converge to homogenous designs.

**Theoretical Clarifications and Formalism.** (Addressing Reviewers VyFH, msgE)
We rewrite **Section 3** to formally define robot morphology as a Kinematic Tree and expand **Section 2.2 and Section 4** to clarify the necessity of evolutionary mechanisms compared to alternatives like GFlowNets, specifically in the context of unconstrained raw XML generation.

**Other Supplemental Experiments and Explanations**
- Experimental Settings: Addressing Reviewer VyFH, we re-conduct the main experiments using strict step limits instead of episodes. The discussion and updated **Table 1** confirms that MorphoGen consistently outperforms baselines under these new settings and the main conclusion remains.
- LLM Selection Analysis: Addressing Reviewer msgE, we expand **Section 5.5** to include Gemini series. The results explicitly demonstrate that performance scales with the model's size and reasoning capabilities.
- Proxy Fitness Effectiveness: Addressing Reviewer LDgZ, we add **Appendix F** to validate our proxy fitness strategy. It facilitates the discovery of morphologies with 28.2% better performance and 96% fewer resources.
- Detailed discussions about necessity of evolutionary mechanisms versus the challenges of applying GFlowNets to unconstrained XML generation, clarifications on the algorithm implementation.
- Visualization: Addressing Reviewer VyFH, we added GIFs of the optimal gaits to our anonymous repository.

We believe these extensive revisions and additional experiments have robustly addressed the reviewers' concerns and significantly strengthen our paper. Thank you again for your time and effort in coordinating this review process.

Best regards, The Authors

---

### Meta-Review · Area_Chair_LMNC · 2026-01-07

**Summary:**

MorphoGen reframes robot morphology optimization as a code‑generation problem, using large language models to directly mutate and evolve raw XML descriptions of robot bodies. Instead of relying on constrained grammar-based search spaces or expensive full-controller training for each candidate, MorphoGen introduces (1) LLM‑driven mutation and crossover on unconstrained XML, (2) a critic LLM that provides textual “gradient-like” guidance, and (3) a hierarchical proxy‑fitness evaluation that enables large-scale exploration. Across four locomotion benchmarks, MorphoGen discovers novel, high-performing morphologies demonstrating that LLMs can effectively navigate vast, combinatorial engineering design spaces.

Reviewers appreciated the core idea of using LLMs to generate robot XML descriptions, the quality of presentation, and the empirical performance.

Several concerns were raised:
- Papers over a year old that had also used LLM-driven evolutionary search for robot morphology synthesis were entirely missed in the submission, taking away a lot of the claimed novelty.
- Evaluation was conducted in relatively simple environments that have been around for a long time, and which were therefore likely not new to the LLM. This calls into question the effectiveness of the approach for genuinely new tasks.
- The final solutions and their quality appears to be tied too closely to the quality of the initial seed population.
- The use of the proxy fitness score which largely relies on policies optimized for other robot designs also appears to hinder large design changes.
- Writing and formalism were found to be imprecise / sloppy in several passages

**Reviewer Concerns:**

Addressed:
- Writing and formalism were found to be imprecise / sloppy in several passages. This is largely addressed.

Partially addressed / unaddressed:
- Papers over a year old that had also used LLM-driven evolutionary search for robot morphology synthesis were entirely missed in the submission, taking away a lot of the claimed novelty. The authors in the rebuttal and paper revision have attempted to rectify this, but in my assessment, properly responding to this requires such a large chagne to the original contribution claims (effectively, "first to consider that robot morphologies could be generated and evolved with LLMs") that it is impossible to achieve within the rebuttal period, and even if it were achieved, it effectively requires re-reviewing the paper as if new.
- Evaluation was conducted in relatively simple environments that have been around for a long time, and which were therefore likely not new to the LLM. This calls into question the effectiveness of the approach for genuinely new tasks. Authors report edit distance relative to previous robot designs for those environments, but the better evaluation here would be to report performance on newly constructed tasks for which working robot designs do not already exist.
- The final solutions and their quality appears to be tied too closely to the quality of the initial seed population. Authors partially address this by acknowledging that the final solutions are indeed significantly influenced by the initial designs, but their sensitivity experiments still operate with only minor perturbations from "expert" designs in the original population.
- The use of the proxy fitness score which largely relies on policies optimized for other robot designs also appears to hinder large design changes.

**Reviewer Scores:**

VyFH: 4 -> 4

msgE: 6 -> 6

LDgZ: 2 -> 2/4

u5DK: 6 -> 6 (they responded explicitly stating this)

Overall, it feels to me that this work is yet to fully respond to the fact that its main claims in the original submission need to be made more precise in response to the pointers to several prior LLM-guided robot design generation approaches. There are also other important concerns pointed out above such as with the evaluation scenarios and the ablation of seed design quality that all need improvements before the core contributions can be said to be properly validated.

---

### Decision · Program_Chairs · 2026-01-26

Reject